# Sensitivity of liquid cloud optical thickness and effective radius retrievals to cloud bow and glory conditions using two SEVIRI imagers

Nikos Benas[1], Jan Fokke Meirink[1], Martin Stengel[2], Piet Stammes[1]

[1]Royal Netherlands Meteorological Institute (KNMI), De Bilt, The Netherlands
[2]Deutscher Wetterdienst (DWD), Offenbach, Germany

*Correspondence to*: Nikos Benas (benas@knmi.nl)

**Abstract.** Retrievals of cloud properties from geostationary satellite sensors offer extensive spatial and temporal coverage and resolution. The high temporal resolution allows the observation of diurnally resolved cloud properties. However,
retrievals are sensitive to varying illumination and viewing geometries, including cloud glory and cloud bow conditions, which can lead to irregularities in the diurnal data record. In this study, these conditions and their effects on liquid cloud optical thickness and effective radius retrievals are analyzed using the Cloud Physical Properties (CPP) algorithm. This analysis is based on the use of SEVIRI reflectances and products from Meteosat-8 and -10, which are located over the Indian and Atlantic Ocean, respectively, and cover an extensive common area under different viewing angles. Comparisons of the
retrievals from two full days, over ocean and land, and using different spectral combinations of visible and shortwave-infrared channels are performed, to assess the importance of these factors in the retrieval process. The sensitivity of the cloud bow and glory related irregularities to the width of the assumed droplet size distribution is analyzed by using different values of the effective variance of the size distribution. The results suggest for marine stratocumulus clouds an effective variance of around 0.05, which implies a narrower size distribution than typically assumed in satellite-based retrievals. For the case with
continental clouds a broader size distribution (effective variance around 0.15) is obtained. This highlights the importance of appropriate size distribution assumptions and provides a way to improve the quality of cloud products in future climate data record releases.

## 1 Introduction

Low warm clouds contribute a large part of the overall cloud effects and feedbacks on the climate system (Zhou et al., 2016).
Forming uniform decks, especially over large oceanic areas around the globe, they increase the planetary albedo and exert a cooling effect in the Earth's radiative balance (Wood, 2012).

Optical and microphysical properties of liquid clouds, specifically optical thickness ($\tau$) and effective radius ($r_e$), are important for the estimation of cloud-radiation interactions and the consequent effects on the atmospheric radiation budget.

They are also used for the calculation of the cloud droplet number concentration (CDNC), which is a key parameter for the assessment of aerosol-cloud interactions (Grosvenor et al., 2018), and the cloud liquid water path (LWP) of marine low clouds, which are a crucial component of the water cycle (Wood and Hartmann, 2006). Furthermore, climate models rely on the measurements or retrieval of these cloud properties for the evaluation of their relevant parameterizations (e.g. Pincus et al., 2012).

These characteristics highlight the importance of continuous monitoring of clouds and their properties, which on a global scale is possible only through satellite observations. In fact, during the last decades, substantial advances have been made regarding the continuous and reliable retrieval of cloud properties. Cloud property data records derived from satellite-based passive visible-infrared (VIS-IR) imagers start already in the early 1980s, based on Advanced Very High Resolution Radiometer (AVHRR) measurements from polar orbiting satellites, e.g. the Satellite Application Facility on Climate Monitoring (CM SAF) Cloud, Albedo and Surface Radiation dataset from AVHRR data - second edition (CLARA-A2, Karlsson et al., 2017) and the Pathfinder Atmospheres - Extended (PATMOS-x, Heidinger et al., 2014) data records; combinations of measurements from polar orbiting sensors, e.g. the ESA's Climate Change Initiative cloud data records, which are based on AVHRR, MODIS, ATSR-2 and AATSR (Stengel et al., 2017); and measurements from polar and geostationary satellites, e.g. the International Satellite Cloud Climatology Project (ISCCP) data set (Young et al., 2018). Additionally, more recent and advanced sensors provide high spatial and/or temporal resolution in more spectral channels, also increasing the number and reliability of cloud properties retrieved. Examples of such sensors include the Moderate Resolution Imaging Spectroradiometer (MODIS) and the homonymous cloud data set (Platnick et al., 2017), and the Spinning Enhanced Visible and Infrared Imager (SEVIRI) and the corresponding Cloud property dAtAset using SEVIRI - second edition (CLAAS-2) data record (Benas et al., 2017).

Cloud optical and microphysical properties are presently routinely retrieved from passive VIS-IR satellite imager measurements, basically following the "Nakajima-King" approach (Nakajima and King, 1990). This retrieval principle is based on the combination of a visible/near-infrared channel in which clouds are non-absorbing and the reflectance is primarily a function of $\tau$ and a shortwave-infrared (SWIR) channel in which clouds are absorbing and the reflectance is primarily a function of $r_e$. Methods utilizing this principle are currently applied to all sensors with an appropriate combination of channels.

Despite the continuous advancements in both satellite sensors and retrieval algorithms, challenging issues remain. One of them is the problematic retrievals reported in liquid cloud optical and microphysical properties, associated with specific illumination conditions (Zeng et al., 2012; Cho et al., 2015; Liang et al., 2015). These conditions include the backscattering directions, where the cloud glory effect is manifested, and scattering angles close to 140°, where the cloud bow effect, which is the equivalent of the rainbow created by cloud droplets, appears (Können, 2017). Retrieval failures and biases in $\tau$ and $r_e$

may occur under these conditions for different reasons and have been reported for cloud glory and cloud bow in MODIS (Cho et al., 2015) and cloud bow in MODIS and the Multi-angle Imaging Spectroradiometer (MISR, Liang et al., 2015), while angular biases under the same conditions were also found in retrievals from Polarization and Directionality of the Earth's Reflectances (POLDER) observations (Zeng et al., 2012).

Another issue in cloud optical properties retrieval, which relates to the cloud glory effect, is the width of the cloud droplet size distribution assumed in the retrieval process. This width is usually represented by the effective variance ($v_e$) of the size distribution or other equivalent measures, e.g. the shape parameter $\mu$ (Petty and Huang, 2011). In case of passive satellite sensors that measure total reflectance, $v_e$ is not retrieved; a constant value is instead assumed and used for the retrieval of $\tau$

and $r_e$ of all liquid clouds. While under most retrieval circumstances the sensitivity of $\tau$ and $r_e$ to $v_e$ is low, this is not the case for special illumination geometries, as was shown e.g. in Mayer et al. (2004) for the cloud glory conditions. Typical $v_e$ values used in satellite-based retrievals lie between 0.10 and 0.15. The former is the value used in MODIS Collection 6, ISCCP-H (Rossow, 2017) and PATMOS-x (Walther and Heidinger, 2012), while in the Cloud_cci data records $v_e$ equals 0.11 (McGarragh et al., 2018). For the CLARA-A and CLAAS records $v_e$ equal to 0.15 is assumed (Karlsson et al., 2013; Stengel

et al., 2014; Karlsson et al., 2017; Benas et al., 2017). In MODIS Collection 5, a standard deviation of a lognormal size distribution equal to 0.35 was used (Liang et al., 2015), which corresponds to a $v_e$ equal to 0.13 (Nakajima and King, 1990). Studies including in situ measurements, however, suggest a significantly wider range of $v_e$ values, depending on cloud types, regions (marine or continental, see e.g. Miles et al., 2000) but also for the same cloud type (Igel and Van den Heever, 2017).

In the present study we analyze irregularities in the diurnal evolution of retrieved cloud optical and microphysical properties ($\tau$ and $r_e$), appearing near the cloud glory and cloud bow geometries, and their sensitivity to the width of the assumed cloud droplet size distribution. For the analysis of the diurnal variability of optical properties, we use data from SEVIRI on board geostationary satellites Meteosat-8 and -10, and the Cloud Physical Properties (CPP) retrieval algorithm (Benas et al., 2017; Roebeling et al., 2006), used in the production of CLARA-A2 and CLAAS-2 data records. Cloud glory and cloud bow, and

the ensuing irregularities in the retrievals, occur in specific time slots depending on the region and the season. Hence, their study is necessarily limited to small areas and specific days, since extensive spatial or temporal averaging would diminish their effects. We focus on two regions and two characteristic days, one over the southeastern Atlantic and the other inland over southeastern Africa, at similar latitudes. Each of these regions is scanned by the two SEVIRI sensors of both Meteosat satellites under different illumination conditions, so that possible effects near the cloud glory and near the cloud bow occur

in the two retrievals at different times. In this way, and by comparing the diurnal evolution of the retrieved optical properties from the two satellites, we can assess the effects of these illumination conditions on the retrievals.

The sensitivity of these effects to the width of the assumed size distribution is analyzed by performing retrievals using different values of the corresponding $v_e$. Intercomparisons of the products derived from these retrievals help in the

assessment of their sensitivity and highlight the importance of selecting the appropriate value of $v_e$. Apart from this analysis, three additional retrievals are performed; one over the same region, but using a different spectral combination of visible and SWIR channels, and two over a land area of southern Africa, using both spectral combinations. Corresponding comparisons of different underlying surfaces (ocean/land) and different channel retrievals provide further insights into the relative

importance of these factors in the retrieval process.

The combined use of retrievals from the two sensors to analyze the effects of cloud bow and cloud glory is based on the assumption that retrievals would otherwise be the same. This is not always the case, since other factors can cause differences between retrievals from the same algorithm applied to the same sensor on different platforms. These factors include, among

others, shadow and other 3D effects, partially cloudy pixels and cloud inhomogeneity, surface effects, and misidentification of clouds (e.g. thin cirrus) or cloud phase. Here we show that the aforementioned assumption is valid in the marine case, while over the land area differences are found. However, the simultaneous analysis from both satellites is still valuable, in terms of verifying common effects from two different points of view.

In the following section we describe the two Meteosat satellites and the CPP retrieval algorithm in more detail, along with the data used and the way these were processed. Section 3 includes the results, focusing first on the retrieval algorithm input and output over the southeastern Atlantic region, their characteristics due to different illumination conditions (Sect. 3.1) and their dependence on the width of the assumed size distribution (Sect. 3.2). Comparisons between retrievals from different spectral pairs and over different underlying surfaces are presented in Sects. 3.3 and 3.4, followed by the discussion and

conclusions.

**2 Data and Methodology**

**2.1 Satellites**

EUMETSAT operates four Meteosat Second Generation (MSG) satellites, namely Meteosat-8, -9, -10 and -11 (also referred to as MSG-1, -2, -3 and -4, respectively), all positioned in geostationary orbit. In September 2016 MSG-1 was nominally

positioned at 41.5° E longitude, covering mainly Africa and the Indian ocean, and in early 2017 the Indian Ocean Data Coverage (IODC) service became operational. MSG-3 was the primary operational satellite for Africa, Europe and the Atlantic Ocean, nominally positioned at 0° longitude, between January 2013 and February 2018. The areas covered by MSG-1 and MSG-3 have a large overlap (see Fig. 1), comprising Africa, Europe, the Middle East and large oceanic regions, which offers new opportunities for synergistic usage of data from the two satellites.

It should be noted that the two satellites deviate from their nominal positions over the course of a day. In the period considered in this study (March 2017), this deviation is most pronounced in the latitude of MSG-1, which ranges between

approximately 5° S and 5° N on a 24-hour basis. This deviation alters the viewing geometry and estimated scattering angles, thus also affecting the retrieved optical properties. To avoid possible consequent misinterpretations, information on the exact position of each satellite, available on a 15-minute time slot basis, was included in the retrieval process.

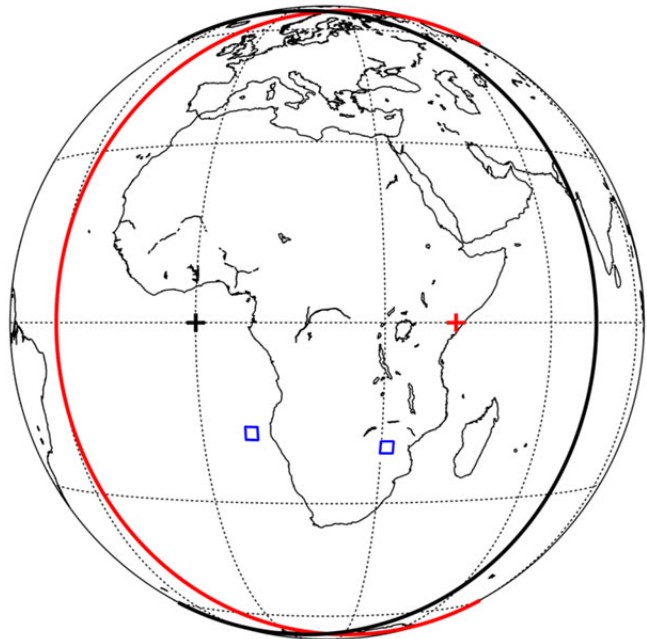

**Figure 1: The overlap area created by the disks of MSG-1 (red line) and MSG-3 (black line). The red and black crosses show the corresponding nominal sub-satellite points. The blue rectangles show the study regions west of the African coast (16.5°-18.5° S, 7.5°-9.5° E) and in the southern parts of Zimbabwe and Mozambique (19°-21° S, 30°-32° E).**

SEVIRI is one of the main instruments on board the MSG satellites. It observes the Earth in 11 spectral channels from the
visible to the thermal infrared and one high resolution broad-bandwidth channel in the visible (HRV), acquiring measurements every 15 minutes at 3 km nadir resolution (1 km for the HRV). To ensure a valid intercomparison between MSG-1 and MSG-3 reflectances and retrievals, calibration of SEVIRI shortwave channels on both satellites was performed using Aqua MODIS Collection 6 reflectances as a reference, instead of the operational EUMETSAT calibration. The approach is described in Meirink et al. (2013) and was extended in this study to include 2017. This yielded calibration slopes
of 0.0267, 0.0229, 0.0235, and 0.0229 mW $m^{-2}$ $sr^{-1}$ $(cm^{-1})^{-1}$ for the MSG-1 0.6 μm, MSG-1 1.6 μm, MSG-3 0.6 μm, and MSG-3 1.6 μm channels, respectively. These values can be compared with the corresponding operational calibration slopes of 0.0241, 0.0233, 0.0209, and 0.0236 mW $m^{-2}$ $sr^{-1}$ $(cm^{-1})^{-1}$, respectively.

## 2.2 Retrieval Method

The CPP algorithm uses measurements from one visible and one SWIR channel to retrieve $\tau$ and $r_e$. For SEVIRI, this is achieved by combining the channel near wavelength $\lambda$ = 0.6 μm with either the 1.6 μm or the 3.9 μm channel (CM SAF, 2016a; corresponding central wavelengths are 0.635 μm, 1.64 μm and 3.92 μm, respectively). CPP requires a cloud mask, and several cloud top properties as input. The cloud mask as well as cloud top height and temperature are obtained using the Satellite Application Facility for Nowcasting (NWC SAF) GEOv2016 software package (NWC SAF, 2016; Derrien and Le Gléau, 2005). The cloud top phase retrieval is based on a modified version of the Pavolonis et al. (2005) algorithm, as described in Benas et al. (2017). In this study only liquid phase clouds are considered. The physical principle of the CPP approach was described in Nakajima and King (1990). CPP is presently used with various satellite imagers, for the simultaneous retrieval of $\tau$ and $r_e$ by comparison with simulated cloud reflectances in the visible and SWIR under different illumination conditions.

For the radiative transfer calculations, a two-parameter gamma size distribution of liquid cloud droplets is assumed, given in Hansen (1971) and also described in Petty and Huang (2011):

$$n(r) = N_0 r^{\frac{1-3v_e}{v_e}} \exp\left(\frac{-r}{r_e v_e}\right) \qquad (1)$$

The constant $N_0$ is provided in Hansen (1971) but is not required here, since the retrieval algorithm is based on normalized quantities. Mie scattering calculations are performed using a Mie code (De Rooij and Van der Stap, 1984), whereby the scattering matrix is calculated and provided in terms of generalized spherical functions. This output is then used as input for the multiple scattering calculations based on the Doubling-Adding KNMI (DAK) radiative transfer model (De Haan et al., 1987; Stammes, 2001), for the simulation of top-of-atmosphere (TOA) reflectances of clouds in a Rayleigh atmosphere for different channels, which are stored in a lookup table (LUT; see below for its layout). The reflectances $R$ are defined as:

$$R = \frac{\pi I}{E_0 \cos \theta_0} \qquad (2)$$

where $I$ is the radiance measured by the satellite in a specific channel, $E_0$ is the downwelling solar irradiance at TOA filtered with the channel's spectral response function and $\theta_0$ is the solar zenith angle. Using radiative transfer calculations for three values of the surface albedo, the  reflectances can be calculated for the actual surface albedo, which is assumed to be constant over ocean (0.05 in the 0.6 and 1.6 μm channels, and 0.02 in the 3.9 μm channel) and obtained from MODIS-based climatologies over land (Greuell et al. (2013) at 0.6 and 1.6 μm and Seemann et al. (2008) at 3.9 μm). The measured reflectances are corrected for absorption by atmospheric gases, of which concentrations are obtained from the European Centre for Medium Range Weather Forecasting (ECMWF) Integrated Forecasting System (IFS) model (water vapour and ozone) or from climatologies (other trace gases). In case of the 3.9 μm channel, the measurement is further corrected for a contribution of thermal emission based on the IFS surface temperature and the retrieved cloud top temperature (CM SAF, 2016b). A match between the measurements and the LUT of simulated reflectances is then sought, yielding the cloud optical

properties $\tau$ and $r_e$. Uncertainties of the retrieved values are estimated based on a 3% relative error in the reflectances. More details on the retrieval algorithm can be found in CM SAF (2016a).

To assess the sensitivity of the optical properties retrieval to the width of the liquid droplets size distribution, multiple Mie and DAK runs were performed, for the creation of seven LUTs. Each LUT corresponds to a different size distribution width, represented by a different value of $v_e$. The seven values of $v_e$ were determined following the selection of Arduini et al. (2005) and their typical reported range (0.01-0.30, see also Miles et al., 2000; Igel and Van den Heever, 2017). Figure 2 shows the corresponding seven size distributions for $r_e$ = 12 μm (Fig. 2a) along with the scattering phase functions for the visible wavelength (0.6 μm) resulting from the Mie calculations (Fig. 2b). Cloud bow and glory features are apparent in all phase functions as peaks near 140° and in the backscattering direction, respectively, but the details of the phase functions for these scattering situations depend on the effective variance. Each LUT contains simulated reflectances at the required wavelengths for various values and ranges of $\theta_0$, the viewing zenith angle ($\theta$), the relative azimuth angle ($\Delta\phi = 180° - |\phi-\phi_0|$), $\tau$ and $r_e$. Table 1 summarizes these LUT characteristics.

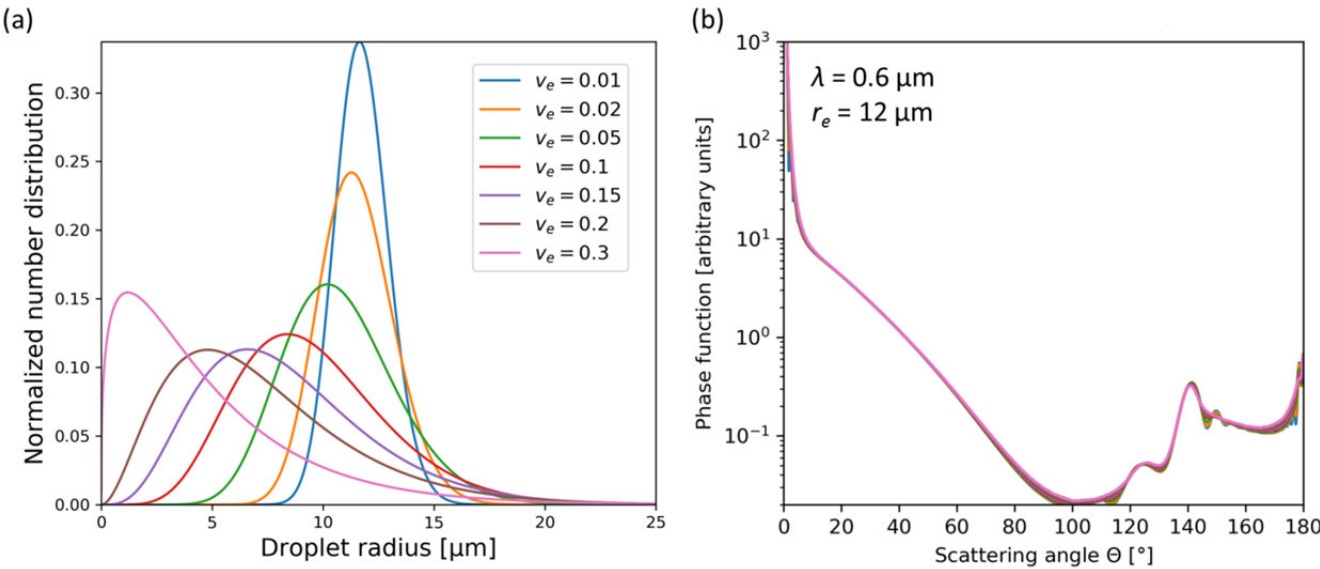

Figure 2: Normalized droplet size distributions for $r_e$ = 12 μm, for the seven values of effective variance $v_e$ used in the Mie and DAK calculations (a), and corresponding scattering phase functions for $\lambda$ = 0.6 μm derived from Mie calculations (b).

**Table 1. Values and numbers of points of the variables comprising the five dimensions of the cloud reflectance LUTs. Each value of effective variance ($v_e$, last row) corresponds to a different LUT.**

| Variable | Values | Number of points |
|---|---|---|
| $\cos(\theta_0)$ | 0.099-1 ($\theta_0$: 0-84.3°) | 73, Gauss-Legendre points |
| $\cos(\theta)$ | 0.099-1 ($\theta$: 0-84.3°) | 73, Gauss-Legendre points |
| $\Delta\phi$ | 0-180° | 91, equidistant |
| $\tau$ | 0 and 0.25-256 | 22, equidistant in $\log(\tau)$ |
| $r_e$ | 3-34 µm | 8, equidistant in $\log(r_e)$ |
| $v_e$ | 0.01, 0.02, 0.05, 0.10, 0.15, 0.20, 0.30 | 7 |

The pixel-based uncertainties were propagated in the calculation of spatial averages, using the methodology described in Stengel et al. (2017) for the propagation of uncertainty estimates from level 2 to level 3 products. This approach assumes bias-free Gaussian distributions for both the retrieved variables and their uncertainties, and estimates the eventual uncertainty depending on the level of correlation among level 2 uncertainties. In our case, while the pixel-based retrieval is performed assuming zero uncertainty correlation between adjacent pixels, some correlation should be expected. Considering this, we used an uncertainty correlation of 0.1 in our estimates, which was also used in Stengel et al. (2017). In order to compare the sensitivity of retrievals to $v_e$ with corresponding uncertainties of these retrievals, we estimated the mean and standard deviation of the assumed Gaussian distributions of uncertainties for the two extreme retrieval cases in terms of $v_e$ ($v_e = 0.01$ and $v_e = 0.30$), and assessed their level of overlap. Hence, lower overlap indicates higher sensitivity of the retrieval to $v_e$.

### 2.3 Selection of study areas and days

Using the different LUTs, liquid cloud $\tau$ and $r_e$ were retrieved from MSG-1 and MSG-3 for two selected days and regions. Specifically, days near the vernal equinox were chosen so that the sun passed over the satellite, yielding glory viewing conditions. Subsequently, two study regions (one over ocean at 16.5°-18.5° S, 7.5°-9.5° E and one over land at 19°-21° S, 30°-32° E, see Fig. 1) were selected based on their high degree of spatial coverage with liquid clouds during specific days in 2017. For the oceanic region the CPP retrieval was performed for March 7, 2017, while for the land region the day selected was March 20, 2017. The CPP retrievals were performed separately for the pairs of channels 0.6 µm - 1.6 µm and 0.6 µm - 3.9 µm. Figure 3 shows the cloud cover of the two areas with liquid clouds during the days selected. Based on the different viewing conditions, and the fact that larger viewing angles lead to larger cloud fractions retrieved, MSG-3 should yield higher cloud cover over the continental and lower over the marine region compared to MSG-1 (see also Fig. 1). This is indeed the case over the continental region, while the good agreement between the two satellites over the marine region should probably be attributed to its almost complete coverage with liquid clouds. The high liquid cloud cover throughout these days ensures the calculation of meaningful statistics of the retrieved cloud properties. In fact, liquid clouds cover more than 80% of these areas during the days selected, including the cloud bow and glory time slots. It should be emphasized here

that all results presented onward were based on pixels where both satellites retrieved liquid clouds, and where $\theta_0$ was within the range defined in Table 1. The latter limitation explains the missing data in early morning and late afternoon.

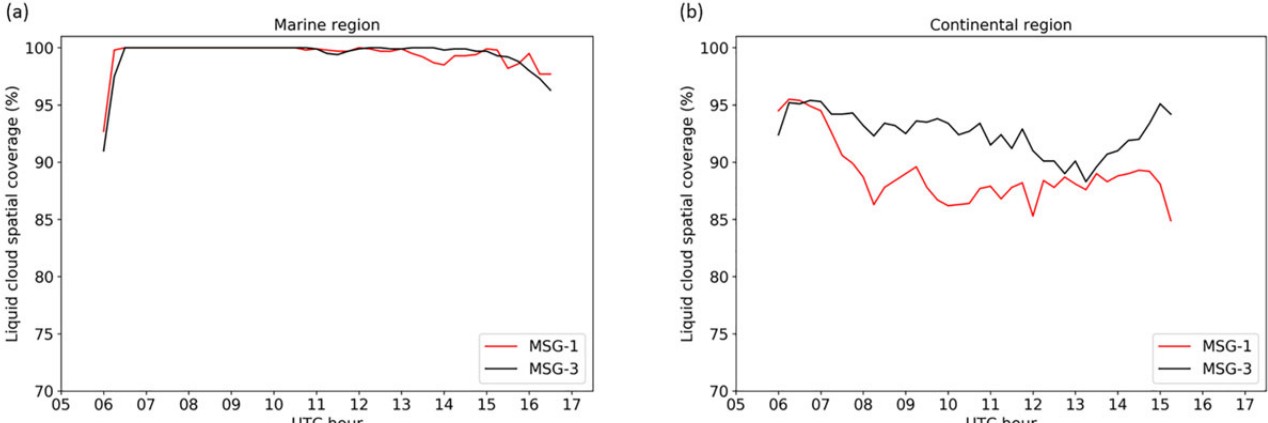

Figure 3: Spatial coverage (%) of the marine (a) and continental (b) regions with liquid clouds during March 7, 2017 and March 20, 2017, respectively, estimated separately from MSG-1 (red lines) and MSG-3 (black lines). The regions are indicated in Fig. 1.

Furthermore, although some afternoon time slots in MSG-1 could possibly be affected by sunglint conditions over the southeastern Atlantic, the good agreement between the two satellites during these time slots ensures that possible sunglint effects do not interfere with the results.

## 3 Results

### 3.1 Irregularities in the CPP diurnal cycle

Figure 4 shows the spatially averaged reflectances of the 0.6 μm and 1.6 μm channels used as input to the CPP algorithm over the southeastern Atlantic on March 7, 2017, separately from MSG-1 and MSG-3. Reflectances were averaged only over pixels with liquid cloud phase retrieved from CPP, to be directly comparable with the CPP output. This output, comprising spatially averaged $\tau$ and $r_e$ values, is shown, revealing a decreasing $\tau$ during the day (Fig. 4d), combined with a relatively constant $r_e$ (Fig.4f). While this decrease in $\tau$ over the region is typical for this marine Sc deck, this is not the case for $r_e$, which typically also decreases (Seethala et al., 2018). Scattering angles, averaged over all pixels in the study region from the two satellites during this day, are shown in Figs. 4a and 4b, along with dotted and dashed vertical lines which highlight the geometries near cloud glory and cloud bow (maximum values and 140° scattering angles, respectively). Scattering angles ($\Theta$) are computed from $\theta_0$, $\theta$ and $\Delta\varphi$ based on:

$$\Theta = \cos^{-1}(\sin\theta_0 \sin\theta \cos\Delta\varphi - \cos\theta_0 \cos\theta) \tag{3}$$

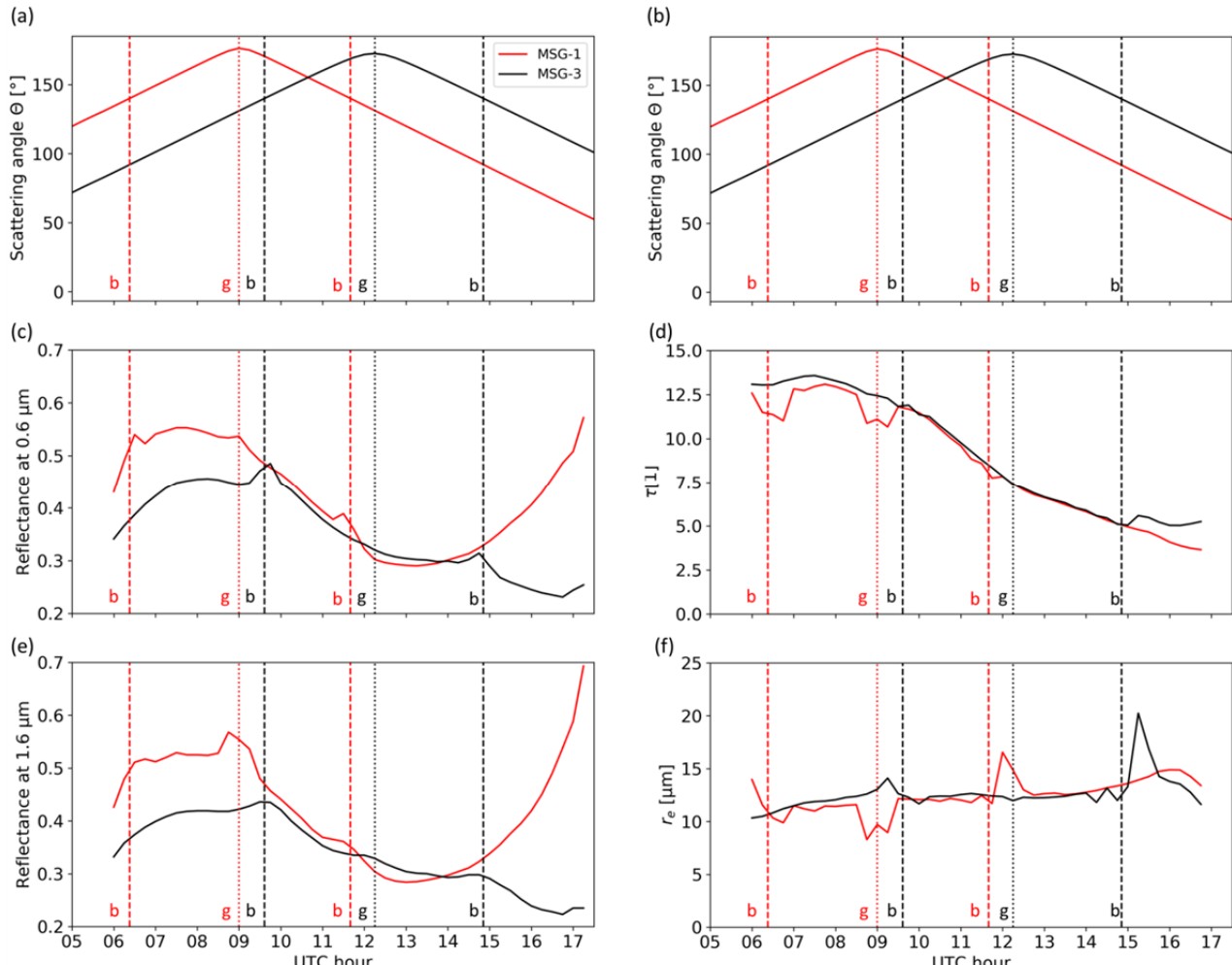

**Figure 4: Diurnal evolution of average cloudy-sky reflectances observed from SEVIRI at 0.6 μm (c) and 1.6 μm (e) and CPP output $\tau$ (d) and $r_e$ (f) over the southeastern Atlantic region on March 7, 2017. Scattering angles (a, b) are shown twice for visualization purposes. All data are shown separately for MSG-1 (red lines) and MSG-3 (black lines). The CPP output is based on retrievals with $v_e$ = 0.15. Dotted vertical lines correspond to the maximum scattering angles, highlighting the cloud glory region, while dashed vertical lines are drawn at 140° scattering angles, roughly the cloud bow regions. Letters "b" and "g", corresponding to cloud bow and glory, respectively, are included next to the vertical lines to facilitate distinction.**

It should be noted that the exact cloud bow angle varies with particle size. Nevertheless, it always lies around 140° (see also Fig. 2b), hence this angle was chosen here for visualization purposes. Furthermore, the glory does not necessarily coincide with the maximum scattering angle depicted in the cases plotted here. However, the angular distance between adjacent time slots (~3°), ensures that cloud glory conditions will occur (perhaps more than once) close to this maximum. The spatial averaging of pixels with slightly different scattering angles in the same time slot also introduces an uncertainty in the value

of $\theta$. For an area size of 2° × 2°, as in our case, the typical range of scattering angle values is about 0.4°. This is narrow enough to ensure no "interference" between adjacent time slots, but also regarding phase function characteristics discussed later. Both cloud glory and cloud bow are apparent as irregularities in the diurnal evolution of reflectances, especially in the visible channel, whereas their effect is partially smoothed in the SWIR. The cloud glory irregularity appears around the maximum scattering angle for that day and region, which is about 176.4° for MSG-1 and 172.6° for MSG-3. Cloud bow irregularities, on the other hand, occur in scattering angles close to 142°. Large discrepancies between MSG-1 and MSG-3 reflectances appear late in the afternoon, with values increasing rapidly for low scattering angles. This difference should probably be attributed to the combined large $\theta_0$ and $\theta$ for MSG-1, but it does not appear to affect the corresponding retrievals.

Despite the possible differences in reflectances measured from the two sensors over the same area and time slot, which should be attributed to the different combinations of illumination and viewing conditions, the retrieval algorithm should in principle compensate for these and ideally produce the same results, which correspond to the real conditions examined from two different angles. In practice, however, this is hardly ever achieved, with many possible reasons contributing to eventual differences, as mentioned in the Introduction. Figures 4d and 4f show the retrieved $\tau$ and $r_e$ separately from MSG-1 and MSG-3. These retrievals are based on $v_e = 0.15$, which is the value used in the CLAAS-2 CPP version. For both satellites, apparent irregularities are centered on the cloud glory in both $\tau$ and $r_e$, with most pronounced discrepancies for $r_e$. It appears, however, that in the cloud bow time slots retrievals are rather normal, with big differences occurring in $r_e$ for smaller scattering angles, namely close to 134°. The very good agreement between the two satellites in other time slots suggests that other factors causing differences in retrievals do not play a substantial role here.

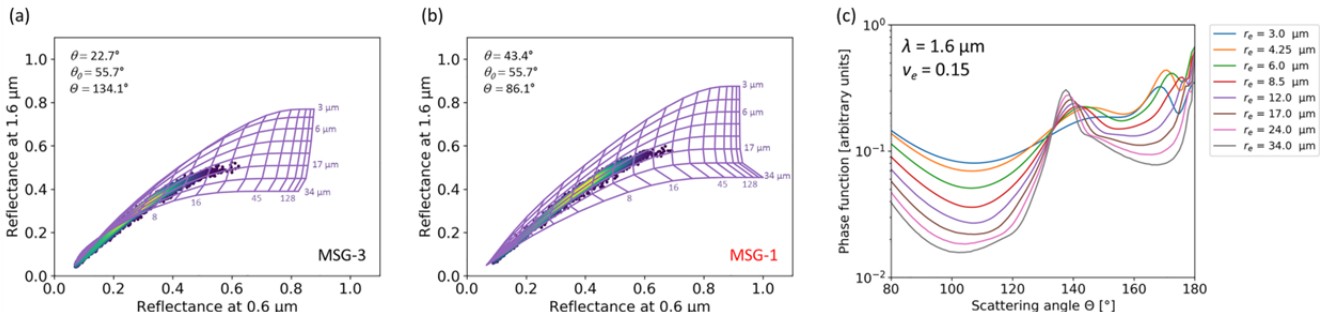

**Figure 5: Density plots of (atmospheric absorption corrected) reflectance observations from cloudy pixels and corresponding retrieval LUTs for the 15:15 UTC time slot in March 7, 2017 over the southeastern Atlantic, separately for MSG-3 (a) and MSG-1 (b). Dark blue to yellow colors show increase in the density of points. (c) Phase functions at 1.6 µm wavelength used in the radiative transfer calculations for the eight $r_e$ values of the LUTs, assuming $v_e = 0.15$.**

Further analysis shows that the cloud bow features are caused by a large number of observations falling outside the LUT, specifically below, leading to $r_e$ retrievals at its highest value (34 μm). This is illustrated in Fig. 5a, which shows the density plot of cloud reflectances observed from MSG-3 at 0.6 μm and 1.6 μm at 15:15 UTC, when $r_e$ peaks (see Fig. 4f), overplotted with the LUT for the same illumination conditions, which was used in the retrieval. For comparison purposes, the corresponding plot is shown for MSG-1 during the same time slot (Fig. 5b). It is apparent that the LUT for MSG-1 covers the observations more adequately, leading to more reasonable $r_e$ retrievals, judging from comparisons with adjacent time slots. The reflectance observed by the satellite is affected by single and by multiple scattering at the same time. Hence, while it is not trivial to find a single scattering signature here, the origin of this LUT inadequacy, occurring for scattering angles around 133°, can probably be traced back to the scattering phase functions used for the LUT calculations. The fact that this LUT characteristic affects optically thin clouds only, where single scattering occasions are more pronounced, supports this explanation. Figure 5c shows the shape of these phase functions in the scattering angle range 80°-180° for all eight $r_e$ values used in the LUT. The overlap of all the phase functions near 133° provides no information on the $r_e$, and leads to the corresponding "collapse" in the left part of the LUT (Fig. 5a). A similar collapse occurs for scattering angles slightly larger than those of the cloud bow but their effect on the averaged retrievals is far less severe. On the other hand, scattering angles in the MSG-1 case lie around 86°, where Figs. 5b and 5c show that $r_e$ is adequately retrievable. These characteristics in the phase functions were also reported for similar scattering angles in the case of MODIS where failure rates also increased (Cho et al., 2015). It should be noted, however, that this inadequacy is characteristic of optically thin clouds only ($\tau$ < 4 in the case of Fig. 5a). It is obvious from the LUT shape in Fig. 5a that for clouds with higher $\tau$, where multiple scattering prevails, $r_e$ can be adequately retrieved.

To quantify the way that specific characteristics of the bulk scattering phase functions affect the failure rates in MODIS cloud optical properties retrievals, Cho et al. (2015) defined the phase function separation index (PS index) as the ratio between the mean and the standard deviation of phase functions at a given scattering angle for all $r_e$ values used in the MODIS LUT. In this way, high values of PS, occurring where phase functions collapse, coincide with high rates of failed retrievals. Following the same method, we estimated separation indices for all phase function groups analysed here, averaging over the $r_e$ values used in the CPP LUT (PS$_r$), but also over the $v_e$ values examined for a given $r_e$ (PS$_v$). This methodology, apart from providing a quantification for the explanation given before, and for similar argumentations given later, offers additional insights regarding angular ranges where retrievals should be expected to succeed or fail. Indeed, the PS$_r$ index corresponding to Fig. 5c, which is plotted in supplementary Fig. S1, confirms that the main issues with retrieval failures should be expected near 133°, with secondarily problematic angles close to 141°, 177° and 180°, very similar to the findings of Cho et al. (2015).

### 3.2 Dependence of retrievals on the size distribution width

A similar analysis in the broader backscattering range (170°-180°) shows that the cause of the irregularities occurring in the cloud glory can be more complicated. While phase function collapses can still occur, it is also known that the shape of the cloud glory depends on $r_e$ and the width of the droplet size distribution, rather than $\tau$. This has already been shown by Mayer et al., (2004) using reflectances at 753 nm and is also verified by our results. In fact, to verify the correct behavior of the CPP LUT in this respect, we examined the LUT reflectances under similar conditions but with thicker clouds (i.e. with multiple scattering prevailing). The southeastern Atlantic region and the MSG-1 observation geometry were selected, whereby both cloud bow and glory effects are apparent in the reflectances (Figs. 4c and 4e), during the same day (March 7, 2017). For each time slot, the viewing and illumination geometry and the $r_e$ were those calculated from the spatial averages of the actual retrievals, while three values of $\tau$ were examined: $\tau = 1$ (very thin cloud), $\tau = 8$ (close to the average retrieved value) and $\tau = 30$ (thick cloud). Using the LUT with $v_e = 0.15$, we plotted the reflectances in the 0.6 μm channel corresponding to these cases. The results (Fig. S2) clearly show how an increased $\tau$ increases the reflectance measured by the satellite sensor. They also confirm that the cloud glory and cloud bow reflectance magnitudes relative to non-glory and non-bow time slots are not affected by $\tau$. As Mayer et al. (2004) nicely described it: "The glory structure sits on top of a multiple-scattering background which of course depends on optical thickness".

Figure 6 shows how the phase function at 1.6 μm changes in the backscattering intensity with varying $r_e$ and $v_e$ (phase functions at both 0.6 μm and 1.6 μm are shown for varying $v_e$). Figure 6a constitutes a zoom-in of Fig. 5c. It shows how the angular distance of the characteristic cloud glory rings, appearing here as local maxima, from the 180° scattering angle, depends on the value of $r_e$ for a given $v_e$. On the other hand, when $r_e$ is given, the width of the size distribution controls the range of these maxima. This is depicted in both Figs. 6b and 6c for wavelengths $\lambda = 0.6$ μm and $\lambda = 1.6$ μm, respectively, and for a typical value of $r_e = 12$ μm and $v_e$ ranging between 0.01 and 0.30. It is apparent in both plots that for narrow size distributions the cloud glory is enhanced.

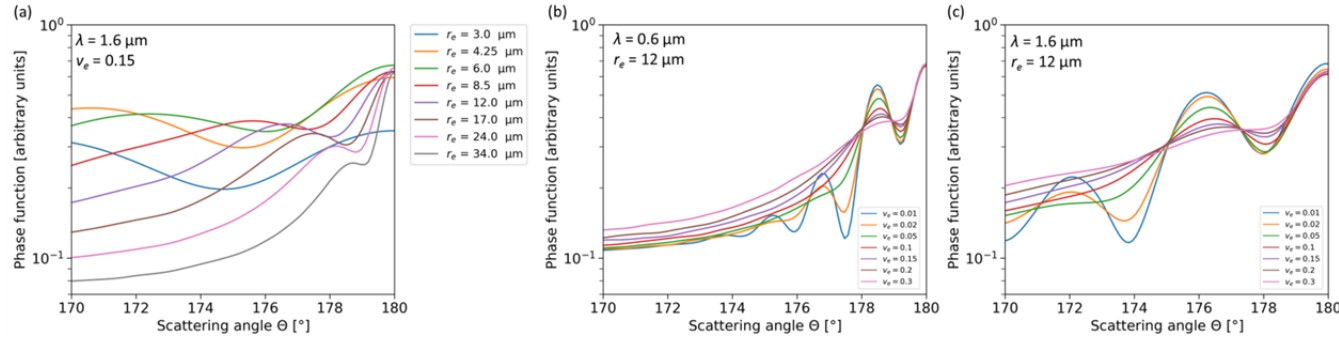

**Figure 6: Dependence of the scattering phase function on $r_e$ and $v_e$ in the backscattering directions. (a) Phase functions at 1.6 μm wavelength used in the radiative transfer calculations for the eight $r_e$ values of the LUTs, assuming $v_e = 0.15$. (b) Phase functions at**

**0.6 μm wavelength used in the radiative transfer calculations for the seven $v_e$ values of the LUTs, assuming $r_e = 12$ μm. (c) Same as in (b), but for phase functions at 1.6 μm.**

Based on the previous analysis, it is natural to examine the CPP output under different assumptions regarding the width of the size distribution and the corresponding value of $v_e$. Figure 7 shows the differences between MSG-3 and MSG-1 $\tau$ and $r_e$

5   retrievals for the seven $v_e$ values examined. In the case of $\tau$ deviations occur only around the glory of each satellite, especially MSG-1, with the diurnal variation appearing smoother for narrower size distributions. These results show that the retrieval of $\tau$ is generally insensitive to the width of the size distribution, except for the cloud glory region. In the case of $r_e$, apart from the variation around the glory, large irregularities appear also near the cloud bow regions (Fig 7b, see also Fig.

10  4f). Due to the distance between the two satellites and the angular distance between cloud bow and glory, which are both close to 40°, the cloud glory from one satellite almost overlaps with the cloud bow from the other, rendering the assessment of the sensitivity to $v_e$ difficult. Furthermore, as will be shown in Sect. 3.4, the assumption that retrievals from the two satellites are the same unless there is a cloud bow or glory condition does not always hold. Hence, the retrievals from the two satellites are re-examined separately, as shown in Fig. 8.

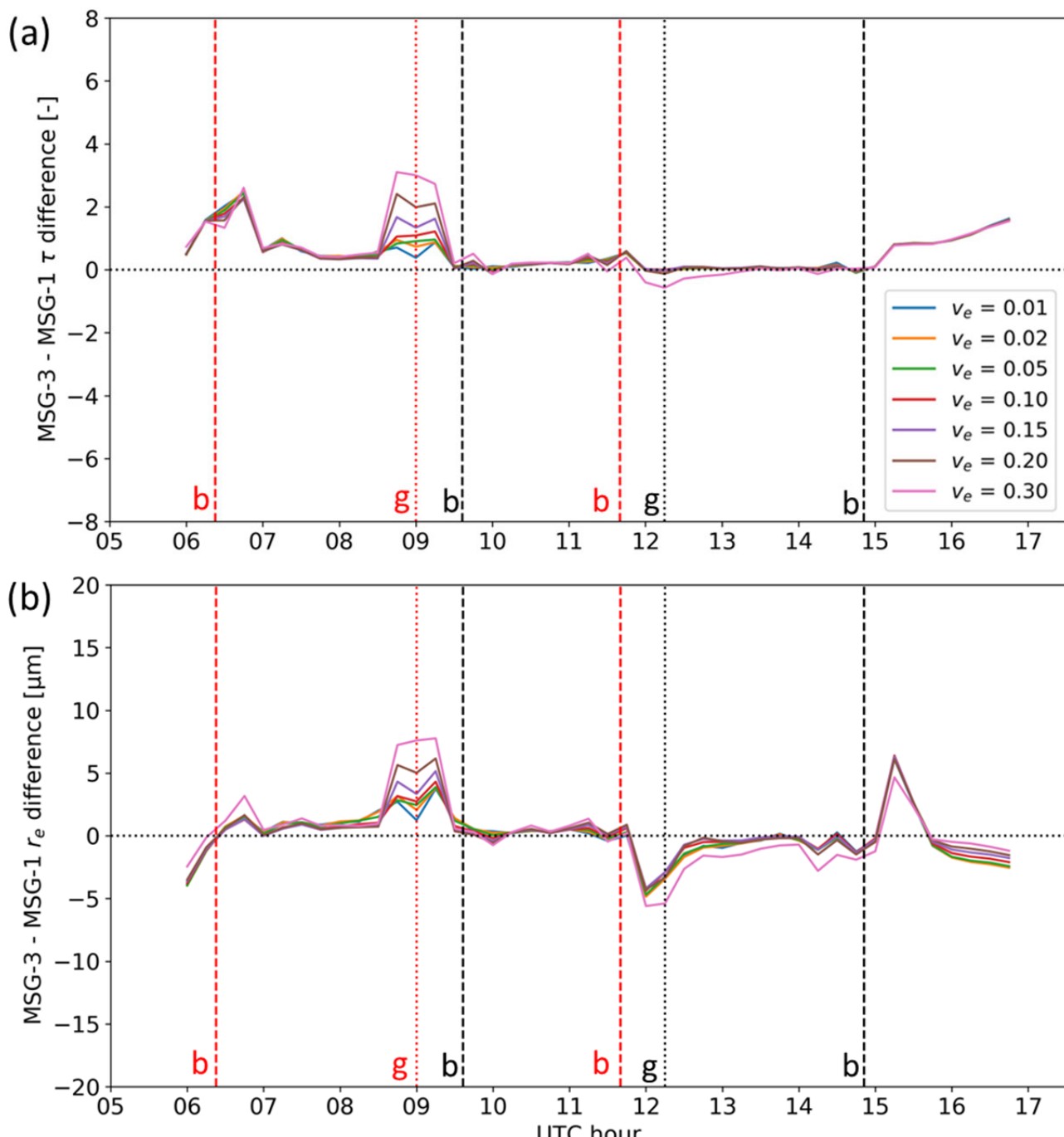

**Figure 7: Differences between MSG-3 and MSG-1 retrievals of $\tau$ (a) and $r_e$ (b) for the seven values of $v_e$ examined, on March 7, 2017, over the southeastern Atlantic. The vertical lines represent cloud glory (dotted, denoted with "g") and cloud bow (dashed, denoted with "b") geometries for MSG-1 (red) and MSG-3 (black), as in Figure 2.**

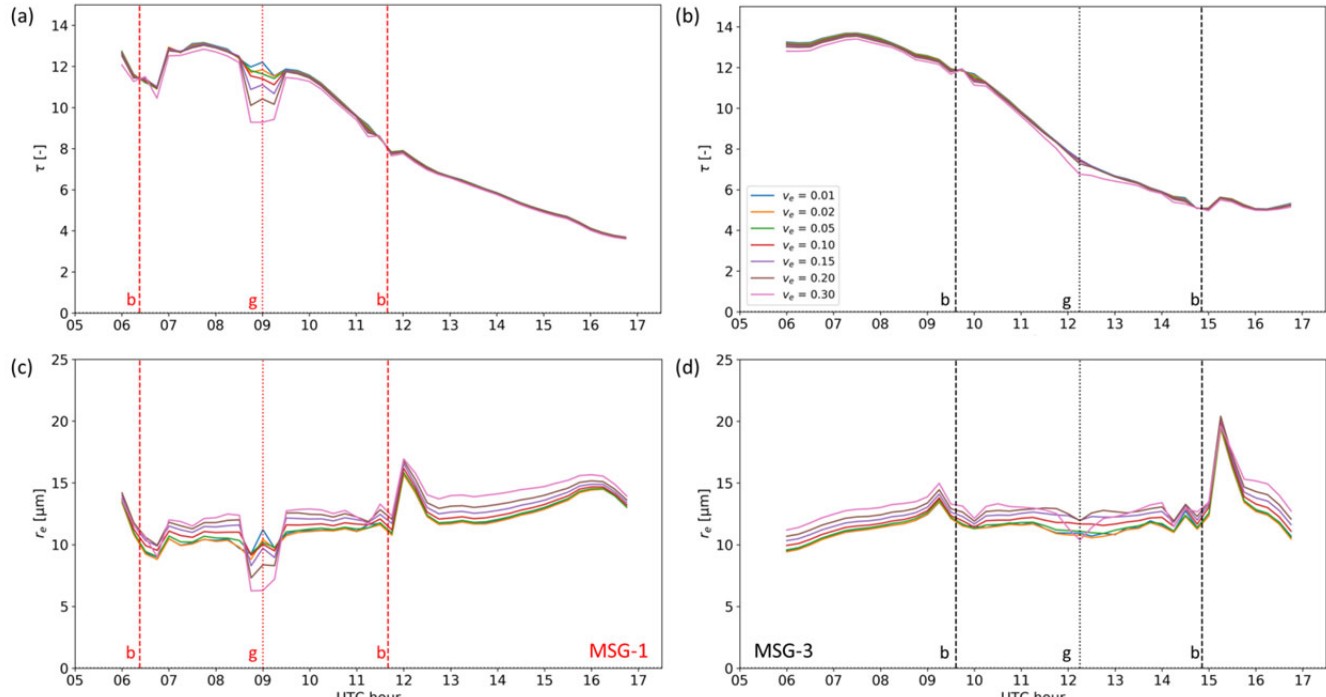

**Figure 8: CPP retrievals of $\tau$ (a, b) and $r_e$ (c, d) based on the 0.6 µm - 1.6 µm channel combination separately for MSG-1 (a, c) and MSG-3 (b, d), for the seven values of $v_e$ examined, on March 7, 2017, over the southeastern Atlantic. The seven $v_e$ values are shown in (b). The vertical lines represent cloud glory (dotted, denoted with "g") and cloud bow (dashed, denoted with "b") geometries for MSG-1 (red) and MSG-3 (black), as in Figure 2.**

It is clear from Fig. 8 that $v_e$ has a strong effect on $r_e$ throughout the day, with differences occurring even during "normal" time slots. Based on the estimated $\tau$ and $r_e$ uncertainties for the two extreme $v_e$ values used, which control the spread of values in Fig. 8, it can be concluded that the difference between the $\tau$ retrievals for the extreme $v_e$ values is smaller than the uncertainty, except in the glory of MSG-1, and the difference between the $r_e$ retrievals for the extreme $v_e$ values is larger than the uncertainty, except in the peak near the cloud bow. The effect of $v_e$ on $r_e$ in the glory is similar to that on $\tau$ under the same conditions, with larger irregularities for wider size distributions being apparent mainly in MSG-1. This difference between the two satellites should be attributed to the different maximum scattering angles: 176.4° for MSG-1 and 172.6° for MSG-3. An inspection of Fig. 6a, and especially the corresponding $PS_r$ separation index (Fig. S3) shows that indeed MSG-1 would be more prone to failures (higher $PS_r$ index values), and thus irregularities, than MSG-3. In the 133° region there is no sensitivity to the size distribution width: all distributions deviate from adjacent time slots. This is because the phase function collapse, shown in Fig. 5c, occurs for all values of $v_e$ used.

Based on the irregularities near the cloud glory shown in Fig. 8 and the logical expectation that $\tau$ and $r_e$ will exhibit a smooth diurnal variation, it appears that narrow droplet size distributions provide outputs that are more consistent with this expectation. This is confirmed by examining the number of pixels which are flagged during the retrieval process, because the pair of VIS and SWIR reflectances lies outside the LUT. CPP provides these flags separately for pixels where the reflectances lie above or below the LUT. Figure 9 shows the percent number of these pixels in the study region separately for MSG-1 and MSG-3 and for flags above and below the LUT.

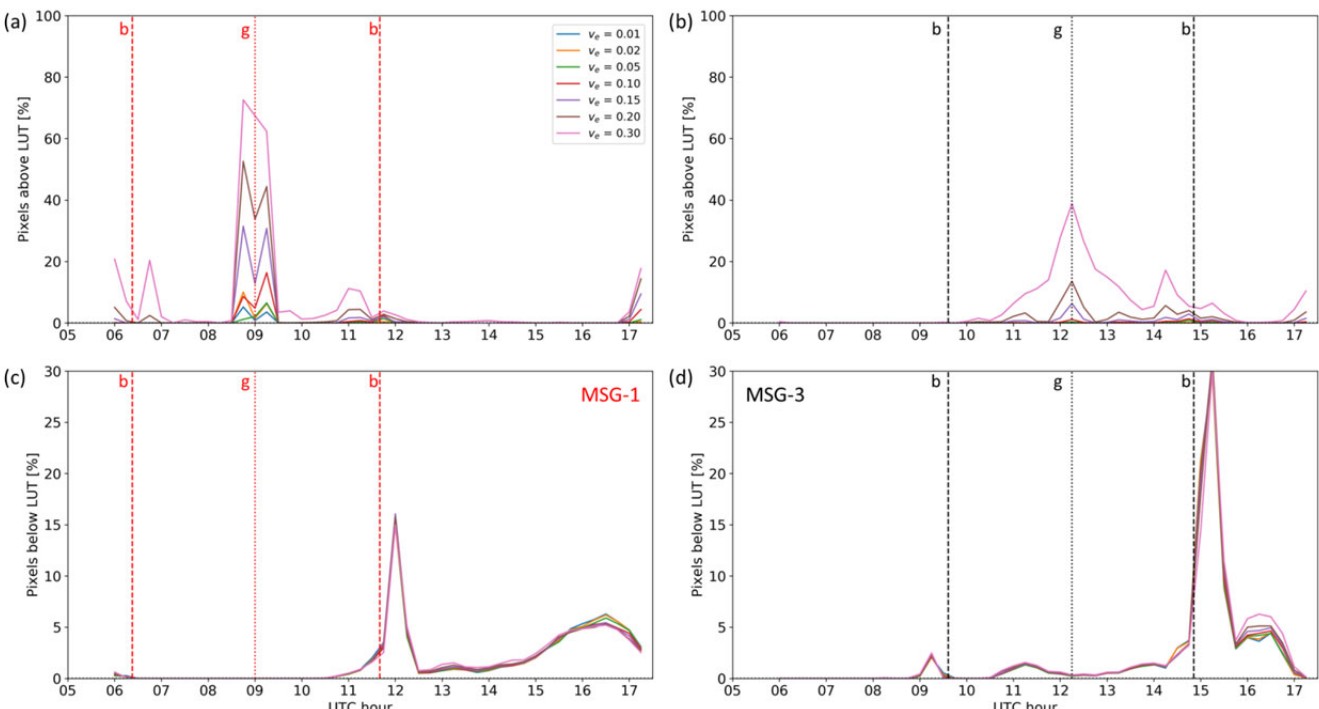

**Figure 9: Fraction of flagged pixels (in %) with pairs of reflectances lying above (a, b) or below (c, d) the retrieval LUT, separately for MSG-1 (a, c) and MSG-3 (b, d) on March 7, 2017 over the southeastern Atlantic. The results are shown for seven LUTs, corresponding to the seven values of $v_e$ shown in (a). The vertical lines represent cloud glory (dotted, denoted with "g") and cloud bow (dashed, denoted with "b") geometries for MSG-1 (red) and MSG-3 (black).**

The number of flagged pixels above the LUT increases rapidly around the cloud glory for wide droplet size distributions, covering up to 40% and 70% of the study region when $v_e$ is higher than 0.15, while it appears that the narrower the size distribution, the less retrieval failures. Distributions with larger widths will have relatively more small particles included (see Fig. 2a). For smaller particles the size parameter ($2\pi r/\lambda$) decreases, moving away from the regime where geometric optics hold, hence in these distributions the cloud glory effect is much weaker, as can be seen in Figs. 6b and 6c. On the other hand, the "collapse" of the LUT which occurs around the cloud bow, due to the overlap of the phase functions, causes failures

below the LUT, of the order of 20% (see also Fig. 5a). Secondary maxima in the flagged "above" pixels (Figs. 9a and 9b) occur for scattering angles slightly larger than 140° and are probably associated with the secondary peak in the $PS_r$ index next to the cloud bow angles (Fig. S1). Note that, contrary to the failures associated with the primary peak of Fig. S1, depicted as pixels "below" in Figs. 9c and 9d, these failures occur above the LUT. Indeed, since the LUT collapses, the

measurements may equally well lie above or below it. It should also be noted here that a comparison of Fig. 9 with Fig. S1 in terms of scattering angles where failures occur, highlights the difference between the failures near the cloud bow and those near the cloud glory: the former should be attributed to the collapse of the phase functions, whereas in the latter, where $PS_r$ index values are much lower, selection of an appropriate $v_e$ value plays the most important role.

### 3.3 Retrievals based on the 3.9 µm channel

CPP retrievals for the same day and region were repeated using the 0.6 µm - 3.9 µm channel combination, instead of the 0.6 µm - 1.6 µm. It is well known that retrievals at the former wavelength are more sensitive to the cloud top compared to the latter, at which the photons penetrate deeper into the cloud (Platnick, 2000). As a result, and because $r_e$ varies vertically, corresponding retrievals are in principle different. Different failure patterns between the two spectral combinations have also been reported, with more successful retrievals for the larger wavelength, which is less prone to failures due to cloud

inhomogeneity (Cho et al., 2015).

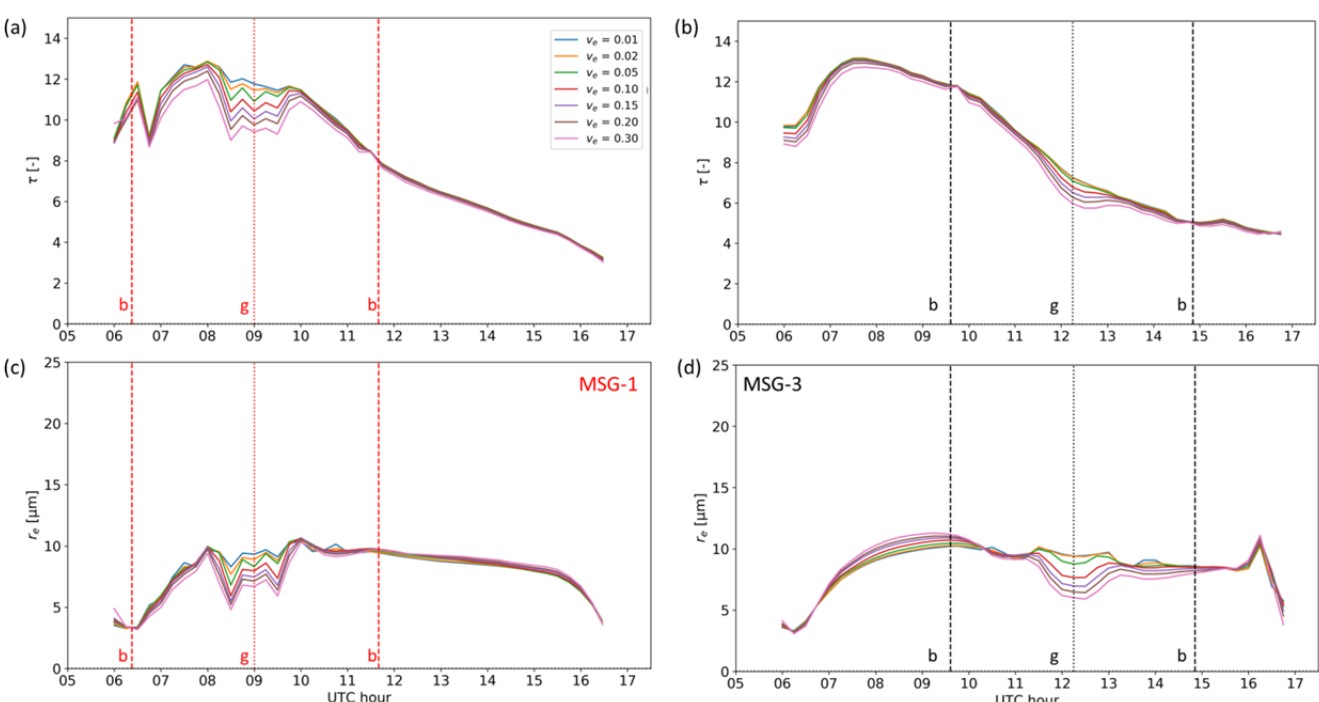

**Figure 10: CPP retrievals of $\tau$ (a, b) and $r_e$ (c, d) based on the 3.9 μm channel separately for MSG-1 (a, c) and MSG-3 (b, d), for the seven values of $v_e$ examined, on March 7, 2017, over the southeastern Atlantic. The seven $v_e$ values are shown in (a). The vertical lines represent cloud glory (dotted, denoted with "g") and cloud bow (dashed, denoted with "b") geometries for MSG-1 (red) and MSG-3 (black).**

Figure 10 shows the diurnal variation of $\tau$ and $r_e$ retrieved using the 3.9 μm channel for the same day and region. Two prominent characteristics are directly distinguishable, compared to the corresponding 0.6 μm - 1.6 μm retrievals. First, a better discrimination of the diurnal patterns around the glory is possible, especially in the case of MSG-3, which is as effective as MSG-1. In fact, in all panels of Fig. 10, the range of values in the glory and adjacent time slots is larger than the

width of the uncertainty distributions. Second, there is no apparent irregularity near the cloud bow, which in the case of the 0.6 μm - 1.6 μm retrieval was caused by the phase functions overlap at 132° scattering angle. The first characteristic stems from the good separation level of phase functions in backscattering angles from both satellites, shown in Fig. 11a and verified by the corresponding PS$_r$ index low values (Fig. S6). It also suggests that in this specific combination of spectral channels and viewing geometry additional information is available regarding the width of the size distribution. In fact, for

scattering angles close to 172°, which is the angle in the glory time slot for MSG-3 in this specific day and region, the single scattering phase functions that correspond to different size distribution widths are much more separated at the 3.9 μm wavelength (Fig. 11b) compared to the 1.6 μm (Fig. 6c), as verified by the corresponding PS$_v$ index values (Figs. S5 and S7). The low PS$_v$ index values associated with this separation level (Fig. S7) hint further to a possibility of $v_e$ retrieval under these specific conditions. The larger time range of retrievals sensitivity to $r_e$ around the maximum scattering angle time slot

compared to the 1.6 μm retrievals should be attributed to the glory features at 3.9 μm phase function which are also more widespread (Fig. 11a). The second characteristic originates in a similar feature, namely non-overlapping scattering phase functions of different $r_e$ values in the 132° scattering angle region for the 3.9 μm wavelength (Fig. 11c) compared to the overlapping phase functions for 1.6 μm (Fig. 5c, see also corresponding PS$_r$ index values in Fig. S8). This feature stems from the fact that for larger wavelengths the cloud bow, which is a geometrical optics phenomenon, is less pronounced, and

renders the 3.9 μm channel more suitable for the retrieval of more realistic diurnal variations of cloud optical properties. Less retrieval failures compared to the 0.6 μm - 1.6 μm retrieval were also found, similarly to the results reported by Cho et al. (2015) on corresponding MODIS channels, although they never disappear completely from the cloud glory time slot. Near the cloud bow  (132° scattering angle), however, they completely disappear.

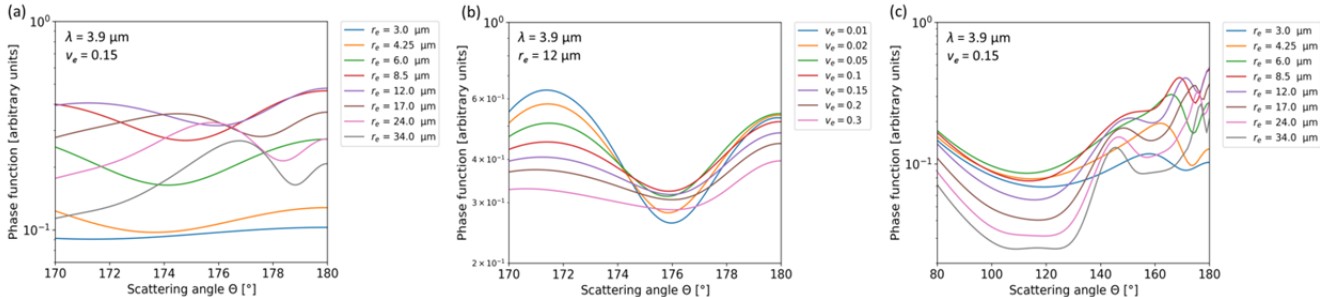

**Figure 11: Dependence of the scattering phase functions at 3.9 μm wavelength on $r_e$ and $v_e$. (a) Phase functions in the backscattering directions for the eight $r_e$ values of the LUTs,, assuming $v_e = 0.15$. (b) Phase functions for the seven $v_e$ values considered, assuming $r_e = 12$ μm. (c) As in (a), but for the scattering angle range 80° - 180°.**

While a direct comparison of $r_e$ values between the 0.6 μm - 1.6 μm and the 0.6 μm - 3.9 μm retrievals should be performed on a pixel basis, the overall smaller $r_e$ values in the latter case hint to the presence of subpixel cloud heterogeneity (Zhang and Platnick, 2011). In fact, based on simulated MODIS retrievals, Bennartz and Rausch (2017) reported that for subpixel fractions of open water above 10% the retrieved $r_e$ at 0.6 μm - 1.6 μm already starts to exceed the one retrieved at 0.6 μm - 3.9 μm. Apart from this, retrieval differences related to imperfect treatment of the 3.9 μm channel cannot be excluded, since this SEVIRI channel is rather broad and requires relatively large atmospheric correction.

### 3.4 Retrievals over the continental region

The results presented so far apply for specific circumstances, namely an optically thin marine Sc cloud over ocean. As previously explained, to examine possible differences caused by different cloud conditions, the same analysis was performed over a continental region, in the southern parts of Zimbabwe and Mozambique (19°-21° S, 30°-32° E, see also Fig. 1). The selection requirements here were also a spatial coverage of at least 80% with liquid clouds only, persistent in most time slots within a day. March 20, 2017 was selected, which is close to March 7, used in the marine case. Combined with the similar latitudes of the two regions, this ensures the presence of similar cloud glory and cloud bow conditions.

Retrievals based on the 0.6 μm - 1.6 μm channels for different values of $v_e$ are shown in Figure 12. Values of $\tau$ reveal an optically much thicker cloud compared to the marine Sc case, with typical values between 20 and 30, increasing even further in late afternoon, while $r_e$ values are also almost double those of the former case. The cloud bow irregularities, especially in the $r_e$, are less pronounced compared to the marine region. This is due to the spectral pairs of cloudy pixels lying in the more orthogonal area of the LUT (see also Fig. 5), thus avoiding the LUT "collapse", which affects thinner clouds. A closer look into the glory area, especially in the $\tau$ case, shows that larger $v_e$ values now provide the smoother diurnal variability. This is consistent with thick continental clouds, for which wider size distributions are expected (Miles et al., 2000). It should be noted that, regarding $\tau$, only in the glory time slots of MSG-3 are the uncertainty intervals for the retrievals at the extreme $v_e$

values non-overlapping. In the case of $r_e$, only in "normal" time slots (neither cloud glory nor cloud bow) are the two extreme values well separated given their estimated uncertainties.

These results are not directly comparable with the marine Sc case: the maximum scattering angles here are 177°-178° for both satellites. Compared to the 172° and even the 176° of MSG-3 and MSG-1 in the marine case, respectively, scattering phase function characteristics at the 0.6 μm channel (shown in Fig. 6b), where $\tau$ retrieval is sensitive, can already differ significantly (see also the corresponding $PS_v$ index in Fig. S4). A similar argument holds for $r_e$ retrieval and corresponding angle and phase function differences in the 1.6 μm channel (Figs. 6c and S5). It should also be noted that, contrary to the marine case, retrievals between the two satellites differ rather substantially in both absolute values and diurnal variability. While 3D effects from the specific cloud type could be causing these differences, the latter were not further investigated, since they do not compromise our results: effects of using different $v_e$ values are still apparent in the individual satellite retrievals.

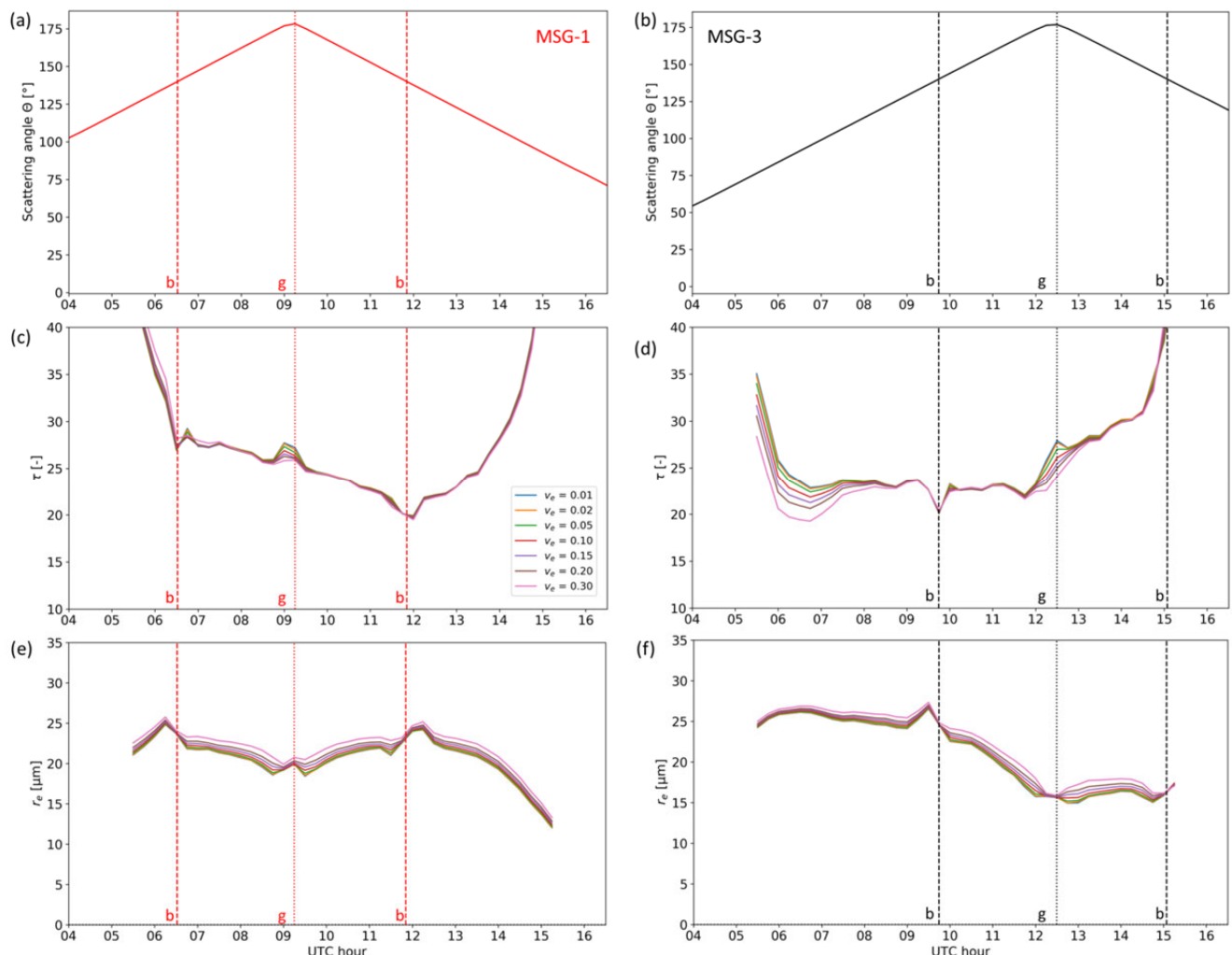

**Figure 12: Scattering angles (a, b) and CPP retrievals of $\tau$ (c, d) and $r_e$ (e, f) based on the 0.6 μm - 1.6 μm channel separately for MSG-1 (left column) and MSG-3 (right column), for the seven values of $v_e$ examined, on March 20, 2017, over the continental region shown in Fig. 1. The vertical lines represent cloud glory (dotted, denoted with "g") and cloud bow (dashed, denoted with "b") geometries for MSG-1 (red) and MSG-3 (black).**

Figure 13 shows corresponding CPP output over the continental region using the 0.6 μm - 3.9 μm channels. As was also implied from the 0.6 μm - 1.6 μm retrievals over the same region (Fig. 12), wider distributions with $v_e$ around 0.15 appear more realistic, with uncertainties in $v_e = 0.01$ and $v_e = 0.30$ non-overlapping in the $\tau$ retrievals around the MSG-3 glory. The value of $v_e$ also appears to affect the $r_e$ retrieval throughout the day: higher $v_e$ lead to higher $r_e$ values, except near the glory region, where this pattern is reversed. For "non-glory" time slots, the range of $r_e$ values found is also larger than the uncertainties. The absence of any cloud bow feature, and the collapse of $r_e$ retrievals near the cloud glory can again be

attributed to corresponding 3.9 μm phase function characteristics in these scattering angles (see also Figs. 11c and 11b respectively).

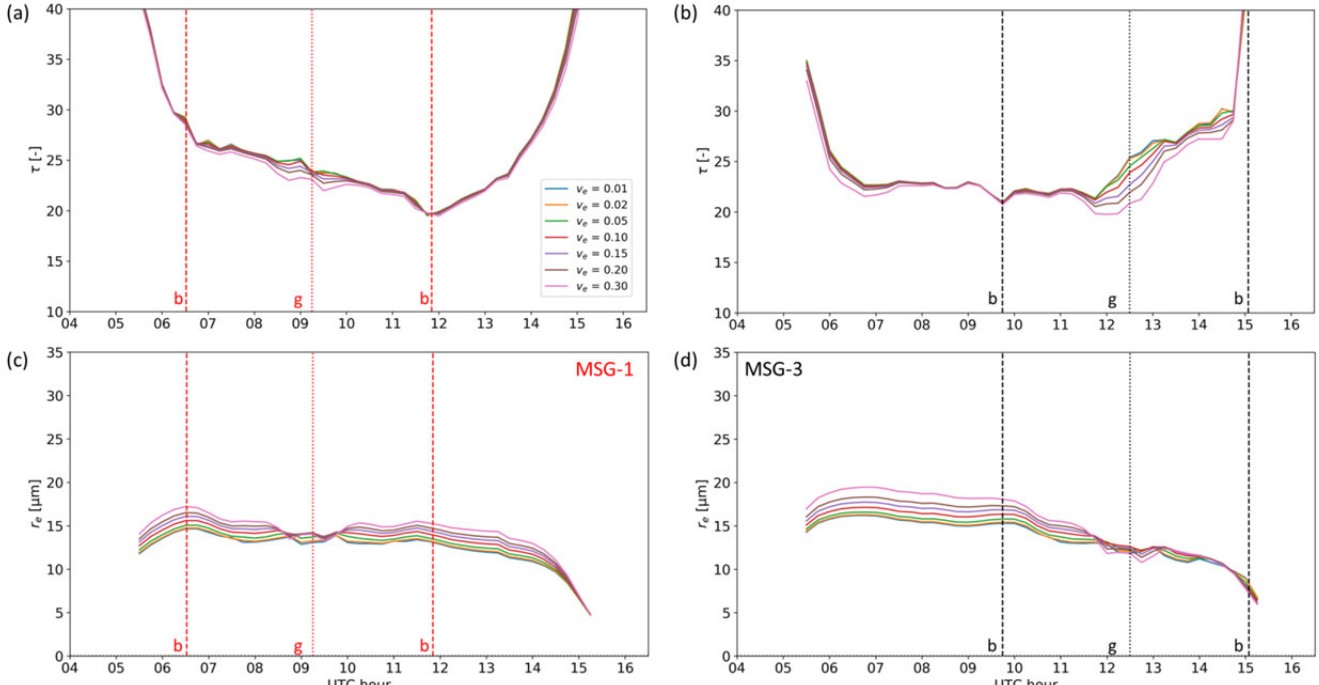

**Figure 13: CPP retrievals of $\tau$ (a, b) and $r_e$ (c, d) based on the 3.9 μm channel separately for MSG-1 (a, c) and MSG-3 (b, d), for the seven values of $v_e$ examined, on March 20, 2017, over the continental region shown in Fig. 1. The vertical lines represent cloud glory (dotted, denoted with "g") and cloud bow (dashed, denoted with "b") geometries for MSG-1 (red) and MSG-3 (black).**

Regarding failure rates in the continental case, it is important noting that in the glory time slot they generally lie below 10%, never exceeding 20% in any channel combination and $v_e$ value, while in the cloud bow they lie around 20% in the 0.6 μm - 1.6 μm retrieval and practically disappear in the 0.6 μm - 3.9 μm, similarly to the marine case. Since the maximum backscattering angles are quite different between the marine and the continental case, decreased numbers of flagged pixels in the latter case might be due to this difference.

## 4 Discussion and Summary

In the present study irregularities in retrieved $\tau$ and $r_e$ from satellite-based passive imagers were investigated using two MSG satellites. The importance of these irregularities is corroborated by the frequency of their occurrence: based on the way scattering angles change during a day, cloud bow irregularities will manifest twice per day in any region. Irregularities associated with cloud glory, on the other hand, require high values of scattering angles. Due to the position of both satellites along the equator, these conditions are met in days close to the two equinoxes. Taking advantage of the large overlap area

between MSG-1 and MSG-3, a marine and a continental region were analyzed under different illumination and viewing conditions. While in principle the common retrieval algorithm should compensate for the different viewing and illumination geometries, and the two satellite products over the same region should agree under any circumstances, monitoring the diurnal evolution of the retrieved optical properties revealed that this is not the case. Results showed that irregularities in this

diurnal evolution are related to scattering phase function characteristics near the cloud bow and cloud glory domains. In the latter case, retrievals were found to be sensitive to the width of the assumed droplet size distribution, expressed by $v_e$. Retrievals based on different SWIR wavelengths also showed that the smaller wavelength (1.6 μm) is more sensitive to cloud bow-induced irregularities than the larger (3.9 μm).

The analysis conducted here raises the question of the most appropriate value of $v_e$ assumed in the retrieval. Measurements from many campaigns have been used for the estimation of the width of the droplet size distribution (see e.g. tables 1 and 2 in Miles et al., 2000 and table 1 in Igel and Van den Heever, 2017). If the corresponding width measures reported in these studies are converted to $v_e$, they lead to a range of values very similar to 0.01-0.30, as was used in Arduini et al. (2005) and in the present study. These results are not contradictory, since different size distribution widths are expected for different

cloud types and under different conditions.

**Table 2. Typical values and ranges of $v_e$ found in observational studies and corresponding review papers.**

| Cloud type | $v_e$ (±1σ) |
|---|---|
| Continental (Miles et al., 2000) | 0.20 ± 0.17 |
| Marine (Miles et al., 2000) | 0.17 ± 0.15 |
| Marine Sc (Miles et al., 2000) | 0.13 ± 0.08 |
| Marine Sc (Mayer et al., 2004) | 0.01 ± 0.002 |
| Marine Sc (Painemal & Zuidema, 2011) | 0.07 ± 0.04 (average profile) |
| | 0.04 ± 0.04 (cloud top) |
| Shallow Cu (Igel & van den Heever, 2017) | 0.09 ± 0.04 |

Table 2 summarizes $v_e$ values obtained from existing observational studies, where different measures of the width of the

droplet size distribution were converted to $v_e$. The $v_e$ values from Miles et al. (2000) and Igel & van den Heever (2017) were based on their tables, in which results from various measurement campaigns are summarized. The continental and marine average values from Miles et al. (2000) are based on all values from their tables 2 and 1, respectively, while the marine Sc average from the same study is calculated based only on the clouds denoted "Sc" in their table 1. Wider droplet size distributions are generally found in continental clouds compared to marine ones. Marine Sc decks exhibit even narrower

distributions. A very narrow size distribution, corresponding to $v_e = 0.01$, was deduced from Mayer et al. (2004) based on aircraft measurement specifically in the cloud glory area, where information on the distribution width is available. Additionally, Painemal and Zuidema (2011), presenting results from a measurement campaign over the Southeast Pacific Sc

deck, report values of the "$k$" parameter, which is an equivalent measure of the size distribution width, varying with cloud height. Specifically, they estimate values of $k$ equal to 0.8 and 0.88 for the average profile and the cloud-top respectively, which correspond to $v_e$ equal to 0.07 and 0.04. Lately, Grosvenor et al. (2018) provided a useful discussion on the effect of the size distribution width on the estimation of CDNC and concluded that a value of 0.10 for $v_e$ is likely to be an
overestimation. More recently, Di Noia et al. (2019) attempted retrieving $v_e$ based on a neural network approach and observations from POLDER-3. Their results show a tendency of the algorithm to also retrieve narrow distributions over ocean ($v_e \sim 0.05$).

The conclusions drawn from the present study are similar, showing that the assumption of narrower distributions, with $v_e$
around 0.05, leads to more reasonable retrievals, at least for the marine Sc cloud type. Instead, a wider size distribution appears more reasonable over the continental region (Sect. 3.4). These differences suggest that, in future retrievals, a cloud type- or region-specific $v_e$ selection prior to retrieval would probably lead to more realistic results under cloud glory conditions. Viewed from the opposite direction, and along with the additional information provided by using different spectral pairs (Sect. 3.3), the results of this study highlight the potential of passive geostationary imagers to retrieve $v_e$ under
specific circumstances. The required information seems to be available in the cloud glory time slot, and a retrieval attempt could be based on an "irregularities minimization" scheme applied to the diurnal variability of the retrieved $\tau$ and $r_e$. Alternatively, apart from the $\tau$ and $r_e$ dimensions in the LUT, an additional $v_e$ dimension could be added, in time slots when corresponding phase functions appear well separated (i.e. near cloud glory conditions). Plans for the next CM SAF CLAAS and CLARA cloud data records include updating the $v_e$ used to a lower value, based on the present results.

**Author contributions**

N.B. and J.F.M. developed the methodology. J.F.M. performed the retrievals. N.B. performed the analysis. All authors contributed in interpreting the results, writing, editing and finalizing the manuscript.

**Competing interests**

The authors declare that they have no conflict of interest.

**Acknowledgements**

This work was performed within CM SAF funded by EUMETSAT.

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
