# Peer review of "Sensitivity of liquid cloud optical thickness and effective radius retrievals to cloud bow and glory conditions using two SEVIRI imagers"

_Atmospheric Measurement Techniques, 2018_

## Referee Comment (RC1) · Anonymous Referee #1 · 12 Feb 2019

Sensitivity of liquid cloud optical thickness and effective radius retrievals to cloud bow and glory conditions using two SEVIRI imagers

Nikos Benas, Jan Fokke Meirink, Martin Stengel, and Piet Stammes

**General comments**

The paper is well written and has an overall clear structure and figures. The topic is interesting and fits well to the aims and scopes of AMT. The authors identified two cloud features, bow and glory in the diurnal cycle of the cloud optical thickness and effective radius of stratocumulus clouds, which caused irregularities and could lead to misinterpretation by the user. The use of the two SEVIRI instruments onboard the Meteosat-8 and -10, which give the stereo perspective, is a great possibility to study these phenomena.

The sensitivity study focused only on the width of the droplet size distribution, which is important parameter and normally fixed for the cloud retrievals. The paper is valuable for people involved in cloud properties determination from image like SEVIRI measurements. The authors suggested two effective variance of the size distribution based on the sensitivity study. The whole study based only on two case studies. It is not shown that these two days are represented for a maritime and continual stratocumulus diurnal cycle, which should be included to make the results useful. Further the authors should discuss if this feature can be flagged for the user and include a suggestion how this cloud specific parameter could be used in standard retrievals as it mention in the abstract, but not discussed later.

In principle the approach the authors present in their paper is very valuable and have great potential to understand cloud glory and cloud bow effects on the diurnal cycle of the cloud properties retrieved from satellite measurements. It is interesting and suitable for publication in AMT. However, I suggest the following revisions:

General comments

**Abstract**

Future climate data record is not mention or discussed in the paper at all, should be add in the discussion summary part.

**Introduction**

p. 3, line 4:

"Another issue in cloud optical properties retrieval, which relates to the cloud glory effects …"

The citation from Mayer et al (2015) should come here already to motivate the sensitivity to ve. ( " While under most retrieval circumstances the sensitivity of $\tau$ and re to ve is low,  this is not the case for special illumination geometries, as was shown e.g. in Mayer et al. (2015) for the cloud glory conditions.")

p. 3, line 24.

This is inconsistent to the abstract the authors mentioned: " .. over different underlying surfaces (ocean/land) .." and here "..over the southeast Atlantic and one characteristic day"

p.3 line 26-28.

Should be move to the summary: " While … properties."

**Data and Methodology**

p.6 line 4-7. "It relies on … illumination conditions." This have not to repeat again. I would suggest to shorten this part.  (see p.2 line 22-27.)

p. 8 line 2. Why only two days? Please discuss.

It would be useful to have RGB images for the two days and two satellite on a time slot.

**Results**

For the spatially averaged reflectances it would be interesting to see the variation (max and min cloud glory and cloud bow effects) and the propagation to the CPP algorithm.

Figure 4: The dotted and dashed line are hard to distinguish. For example, dotted dashed line and dotted line, should be better.

p. 14, line 13. Expression  "more natural output" , please rephrase.

p. 15, line 8. "this does not necessarily mean that the actual droplet size distribution is so narrow." Should be discussed how this could be verified in the conclusion.

**Discussion and Summary**

The authors should more discuss if this feature can be flagged for the user and include a suggestion how this cloud specific parameter could be used in standard retrievals as it mention in the abstract.

p. 19, line 12. How often do these irregularities happen?

Should be critical discussed that the finding depends on the optical thickness of the cloud and the cloud types.

**Typo**

p.4 line 9: Data and Methodology

---

## Referee Comment (RC2) · Anonymous Referee #2 · 14 Feb 2019

Cloud properties retrieved from data of geostationary satellite sensors are largely employed by the scientific community. However, retrieval failures, i.e., the pixels implicitly excluded from the retrieval population could lead to biased estimates. Thus, it is of upmost importance to quantify failures, understand their sources and, if possible, develop algorithms of reprocessing of excluded pixels. Retrieval failures arise for various reasons. Some failure types were addressed in the cited in the manuscript literature. The work under reviewing is devoted to observation conditions favorable to cloud bow or cloud glory. The main idea of the work is to compare retrievals from the same type of data (SEVIRI reflectance) and for common areas but under different viewing angles. That idea is fruitful and leads to convincing results. The work under reviewing

addresses relevant scientific questions it is within the scope of AMT. I recommend that the paper be published in AMT after minor revisions.

Specific comments:

Figures 6 and 11 show phase functions at intervals of angles of cloud bow and glory. It is seen that the phase functions are very sensitive to the effective size and to the effective variance. The reflectance of a cloud as measured from satellite at the top-of-atmosphere should be less sensitive because of multiple scattering. It would be instructive to provide figures of angular dependence of the reflectance as additional figures along with Figs 6 and 11. Additional figures can be done for one typical value of the cloud optical thickness, say 8, using the LUTs created by the authors. One observation geometry, say MSG-3 and the region west of the African coast, will be enough. Some of the Sun-satellite geometry angles can be constant and correspond to the case of March 7, 2017. The geometry angles should vary so that the intervals of scattering angles correspond to the intervals angles of Figs 6 and 11.

Figures 12 and 13 show retrievals results for 0.6 $\mu$m - 1.6 $\mu$m channels and 0.6 $\mu$m - 3.9 $\mu$m channels separately. There is difference in the retrieved values, especially for the effective size. Sensitivity to the effective variance is seen as well. The authors should discuss how those properties can be used to estimate the effective variance.

Technical corrections.

Page 3 line 16 Replace Mayer et al. (2015) by Mayer et al. (2004).

The dotted and dashed lines are hardly distinguished in figures. I would prefer to see straight vertical lines of different color.

---

## Referee Comment (RC3) · Anonymous Referee #3 · 15 Feb 2019

**General comments**

The authors describe two diurnal cycles of liquid cloud properties, optical thickness $\tau$ and effective radius $r_e$, as derived from the CPP algorithm using data of two MSG satellites, Met-8 and Met-10, that observe the same area at the same time from two different positions. The paper addresses the question whether particular scattering geometries, cloudbow and glory, affect the observed cloud properties. Through the peculiarities of the phase functions for these angles the derived properties might be biased w.r.t. other "normal" scattering geometries and lead to spurious results, i.e. in the evaluation of diurnal cycles. To investigate this question the authors employ two SEVIRI instruments

that operate from two different positions and apply a Nakajima-King method to two pairs of satellite channels (0.6-1.6 $\mu$m and 0.6-3.9 $\mu$m). Furthermore, they assess the effect of the effective variance $\nu_e$ of the particle size distribution of the results and hint at a possibility to improve the determination of optical thickness and effective radius. The paper is well structured and addresses an interesting topic that fits to AMT. However, I find that many uncertainties remain and some discrepancies in the results should be addressed and explained in a more direct way. Therefore, the manuscript should be published in AMT after major revisions.

**Specific comments**

The paper combines methodologies mainly from Arduini et al. (2005) and Cho et al. (2015) with the results of Mayer et al. (2004).

The main deficiencies of the paper:

1. The authors talk about the cloudbow effect for scattering angles close to 140° (page 2, line 32) and the "collapse" of the phase function at a scattering angle of 132° and cite to this end the results of Cho et al. (2015) about retrieval failures for MODIS. Similarly, they cite this paper for the glory effect that also leads to retrieval failures for MODIS, but they also mention the work by Mayer et al. (2004) that uses the feature of the liquid cloud phase function around the backscatter direction to retrieve optical thickness, effective radius and effective variance of the particle size distribution. I think that in this paper two different effects are mixed together: on one side, the "bumps" in the phase function observed at the bow and glory geometries; on the other side the reduced sensitivity of the observations to effective radius due to the "collapse" of the phase function close to the same observation geometries.

   An increased phase function intensity for particular scattering angles has as a consequence "stripes" of higher reflectivity for these scattering angles (as observed by Mayer et al. (2004) in the glory region) that, on their turn, could have as a consequence artificial "stripes" of higher optical thickness when the underlying LUT in the Nakajima-King retrievals does not consider these features (e.g. because of an unsufficient angular sampling of the phase function).

The "collapse" of the phase function instead produces retrieval failures due to the fact that the LUTs are narrower, especially for thin clouds, such that the retrieval is more prone to failure because the retrieval/radiative transfer/calibration uncertainties can "push" the observation out of the LUT more easily (cited more or less literally from Cho et al. (2015)). However, this collapse takes place in this paper (Fig. 5) at 132° and *is* the cloudbow effect found in Cho et al. (2015). The authors explain this effect in a nice way, but they assert that "in the cloud bow time slots retrievals are rather normal, with big differences occurring in $r_e$ for smaller scattering angles, namely close to 132°" (page 10, line 15-16). Thus, they mix up these two aspects that shall be separated clearly in the revised version of the manuscript. Correspondingly, it shall also be considered to plot the "collapse" angles instead of the cloudbow angles.

As far as the glory is concerned, I would suggest not to talk about maximum scattering angles and label them as "glory" (page 18, line 6) but to check the behaviour of the phase function for angles larger than say 170° and then argument whether a particular effect on the cloud retrieval is expected at all (and in case which effect) or not. In this sense I think that also the information about the 0.6 $\mu$m phase function used in the argumentation on page 18, line 5-10 should be shown. Furthermore, the same effect in this scattering angle range (glory) might be expected as for the cloudbow as is explained in Cho et al. (2015). The "collapse" of the phase function mentioned at 132° is the most clear one, but Cho et al. (2015) identify further angles where this reduced sensitivity is also observed: they find 133°, 142° and 177° for their MODIS example. Even if at slightly different angles, this effect is present in this manuscript as well (see Fig. 5c and Fig. 6a) and could
be the reason for the irregularities investigated. In particular, the 177° angle is a "glory angle" such that retrieval failures in this scattering angle range might also be traced back to a "phase function collapse".

These aspects shall be addressed in more clarity and the analysis adapted.

2. To observe the effects described above I find the approach of using two fixed regions where the results of $\tau$ and $r_e$ are averaged (I think at least they are averaged: page 9 line 3-4 is not completely clear in this respect) to be not optimal since it might wash out the effects. In fact, all these "ripples" in the phase function take place in few degrees such that the approach of using regions with 5°×5° or 4°×4° size may complicate the identification and explanation of the related retrieval effects. In this sense, do for instance all pixels at 9 UTC in Fig. 4 have maximum scattering angle? I think that it should be discussed how strong the scattering angle can vary inside a SEVIRI pixel and inside the regions investigated in order to assess whether the expected effects can be identified or to what extent they are weakened by the averaging procedure. Furthermore, for a clearer illustration of the results I think that an additional picture showing one area (i.e. a 2D plot in latitude and longitude) in the cloudbow and/or glory slot would help interpreting the results (similarly to Fig. 10 in Cho et al. (2015)).

3. Fig. 6b shows that there are angles of reduced sensitivity to the effective variance while most of the phase function shows a clear dependence on $\nu_e$ in this angle range. Can you see this "collapse" of the phase function w.r.t. $\nu_e$ in Fig. 7? Does this dependency of the phase function on $\nu_e$ extend to other scattering angles as well? Looking at Fig. 2 it would be interesting to shortly explain where an effect due to the dependence of the phase function on $\nu_e$ can be expected.

4. It has never been clearly stated in the manuscript whether the "flagged pixels" (e.g. page 14, line 14) contribute to the plots (e.g. Fig. 4). However, page 10 line 24-25 suggests that the $\tau$-$r_e$ results for the flagged pixels are used for the

statistics (i.e. the diurnal cycles). From my point of view, these flagged pixels correspond to the failures investigated in Cho et al. (2015) and thus should not be used.

Similarly, do you show in Fig. 4c,4e the mean reflectance of the box or the mean cloud reflectance? This uncertainty arises from the phrasing "used as input to the CPP algorithm" (page 9, line 2) which seems to imply that you mean the cloudy pixels alone, since no retrieval is run for cloudfree pixels, I suppose. And what about the scattering angles/optical thickness/effective radius?

5. In Fig. 4, the cloud glory regions (red and black) show different behaviours: the red one (MSG-1) shows a strong irregularity made up of two strong local minima and one local maximum in the optical thickness plot (d), while the black one (MSG-3) shows a weak local mimimum alone. Please explain why there is this difference.

6. I am missing an overall discussion about the plausibility of the retrieved diurnal cycles. This would increase also the plausibility of the investigations shown in the entire paper. For instance:

   - Can one expect that marine Sc has an almost flat $r_e$ diurnal cycle (Fig. 4) while the optical thickness is decreasing strongly, a hint that the thermodynamic conditions the clouds are developing in are changing during the course of the day?

   - The diurnal cycles in Fig. 4 and Fig. 10 differ: the 3.9 $\mu$m retrieval produces a lower $\tau$ and a lower $r_e$, although in an adiabatic environment one would expect higher $r_e$ at the cloud top, where the 3.9 $\mu$m retrieval is more sensitive. If you think that "subpixels fractions of open water" (page 17 line 9-10) are the reason for this, you might take a look at the HRV channel, if it is available over these regions, for a first check about this hypothesis. What is

the uncertainty of the 3.9 $\mu$m retrieval, which should be higher than the one for the 1.6 $\mu$m?

- MSG-1 and MSG-3 in Fig. 10 provide different diurnal cycles of $r_e$. While MSG-1 seems to observe a decrease in $r_e$ at around 9 UTC, MSG-3 yields an increase and a decrease afterwards.

- In Fig. 12 MSG-1 and MSG-3 also provide very different diurnal cycles: not only the absolute values but also the variations in time are different, both in $\tau$ as well as in $r_e$. How can this be explained? Such strong differences preclude of course the use of simultaneous observations of the two satellites, both for physical/meteorological investigations of cloud properties and for the purpose of the present paper.

7. The continental case is said to be "not directly comparable with the marine Sc case" (page 18, line 5). If this case is shown, and I think the paper benefits from this since it shows a cloud with higher optical thickness and much higher effective radius, it should be done in more detail: see my comment above about the phase function and the explanation of the diurnal cycles. Further aspects that are not completely clear for this case are the fact that $\tau$ shows a dependency on $\nu_e$ while $r_e$ shows none in the glory geometry, and the short temporal dispacement between the small local minima in $\tau$ for the cloudbow and the local maxima in $r_e$ (Fig. 12).

8. Retrieval results still seem to show a relatively small variability w.r.t. $\nu_e$. Is this sensitivity to $\nu_e$ comparable to the retrieval uncertainties or retrieval errors or is it larger?

9. The manuscript demonstrates that particular geometries like the cloudbow and the glory can lead to biased optical properties, but what would you propose in order to reduce this bias, keeping in mind that in 15 minutes (from one slot to

the next) also cloud physics can vary (the cloud can thin out, become thicker, its particle size distribution can change...)?

10. The paper should shortly discuss/mention at one place the reasons why the same retrieval from two satellites at the different locations could yield different results, apart because of glory and cloudbow. Here I think of shadow effects, partially cloudy pixels, cloud inhomogeneities, 3D radiation effects, surface BRDF, mixed phase clouds, misidentification of thin cirrus on top... This is the basis for the synergistic use of the two MSG spacecraft. Parts of this discussion are e.g. at page 8 line 8 and page 10 line 11.

Further comments:

**Title**: Since the paper presents results for two selected days over two selected regions I recommend to add "A case study" somewhere to the title.

**Abstract:** Please mention that you analysed two days of data.

**Figures:**

- I suggest to merge Fig. 3a with Fig. 4 and Fig. 3b with Fig. 12.

- Please use the same colors for MSG-1 and MSG-3 in all figures.

- Please add the solar zenith angle to Fig. 4 in order to understand when $\theta_0$ is reaching $90°$, i.e. sunrise and sunset. This might explain for instance the increasing reflectance at $0.6\,\mu$m in Fig. 4c and 4e.

- I suggest to move Fig. 5c to Fig. 6.

- For all figures with glory and cloudbow: it would be helpful for the reader to write directly into the plot which vertical line is glory and which one is cloudbow.

- For all figures with diurnal cycles: it would be easier for the reader if each panel contained MSG-1 or MSG-3 somewhere to distinguish the satellites at a glance.

- Since only hours are used in the diurnal cycle plots I think that e.g. "05" or "5" would be better than "0500".

- Units should be expresses either as e.g. "Θ / degree" or "Θ [degree]" but not "Θ (degree)". Furthermore "Reflectance (0.6 $\mu$m)" should read "Reflectance at 0.6 $\mu$m" with no unit. Instead of "Hour (UTC)" I suggest "UTC hour".

- Fig. 5 is too small. Furthermore, it is probably not a "scatter plot" (page 10, line 25-26) but I guess a 2D histogram. In that case the colors should be explained as well.

- Please add a (dotted) line at height 0 ($r_e$ or $\tau$ difference = 0) in Fig. 7.

**Page 4, line 21**: I cannot believe that the MSG-1 satellite is moving so fast and so much (10° latitude in 24 h) around its subsatellite point. Please check this in more detail! This could have an important impact of the observation geometry.

**Page 6, line 21**: Why do you need three values of the surface albedo?

**Page 6, line 24-27**: Please give a reference (or a short explanation) for the gas absorption correction and the thermal emission consideration.

**Page 7, line 3-4**: The size distributions do not depend on wavelength (line 3), but the phase functions do (line 4), so please shift "for the visible wavelength (0.6 $\mu$m)" after "phase functions". Please correct also the caption of Fig. 2.

[Figure]

**Page 7, line 4-5**: Please indicate in the text and/or in the figure where the cloudbow and glory features can be observed.

**Page 7, Table 1**: Is there such a set of LUTs for every value of the surface albedo mentioned in the text? Please explain this.

**Page 8, line 4**: Please indicate in the text here and not only in the caption of Fig. 1 the details of the region coordinates.

**Page 8, line 5**: Please mention which quantity has been used to assess "uniformity" of the cloud deck.

**Page 8, line 8-10**: This argument, related to the different viewing conditions, also depends on the cloud field observed. If the Sc has dimensions that are anyway smaller than the spatial resolution of SEVIRI, only small differences might appear here.

**Page 8, line 10**: What does it mean that MSG-1 detected "more ice clouds"? Are there ice clouds during these days? Do they contribute to the cloud cover shown in Fig. 3? Are there ice clouds that MSG-3 does not detect and contribute to the retrieval results (Fig. 4 onward)? Are ice clouds maybe one of the reasons for the differences in cloud cover from MSG-1 and MSG-3? Which further factors might explain these differences?

**Page 8, line 14**: At this point of the manuscript it is not clear yet when the cloudbow and glory geometry occur, so please explain this in the text. Nevertheless, if you merge this figure with Fig. 4 or 12, as suggested above, this remark is superfluous.

**Page 8, line 21**: In the MSG-1 curve in Fig. 3a there are discontinuities in the afternoon while the MSG-3 cloud cover is very smooth. Might they be caused by sunglint? Which possible effects might explain these differences otherwise?

**Page 10, line 6-7**: "with values increasing rapidly ..." → Please explain.

**Page 10, line 10**: "different illumination conditions" → please explain.

**Page 10, line 27**: Please mention the observation conditions that are shown here.

**Page 11, line 1-3**: This is an interesting point and also not obvious since the reflectance observed by the satellite is affected by single and by multiple scattering at the same time. Thus it is not trivial to find a signature of the single scattering properties in this quantity. Please consider mentioning this aspect in the text.

**Page 11, line 9**: Please quantify "thin".

**Page 11, line 15**: Is this assertion from Mayer et al. (2004) who used reflectances at 753 nm also valid for other wavelengths? I think you can cite your plots as well to explain this.

**Page 12, line 9-11**: Could you please explain what you mean with this sentence, in particular with "their differences"?

**Page 12, line 11**: What is meant with "This is due..."?

**Page 12, line 12-13**: If you "give up" your synergistic approach of using MSG-1 and MSG-3 you might consider showing results from only one satellite. This would make the next figures "lighter" and you can eventually mention that these results are confirmed (not shown) by the other satellite.

**Page 13, Fig. 7**: Why is the effect of the glory smaller for MSG-3?

**Page 14, line 7**: Please specify "significant effect" and put it in relation to the uncertainty of the $r_e$ retrieval.

**Page 14, line 8**: Please explain what you mean by "The effect on the glory is similar to the $\tau$ case".

**Page 14, line 14**: "which are flagged" as bad quality? As uncertain?

**Page 15, line 10-11**: "... these distributions cannot capture the cloud glory adequately."

Do you mean that such distributions do not show the glory effect or that Mie theory is not adequate for such distributions? The size parameter for 1 $\mu$m particles (small cloud droplet) at 1.6 $\mu$m is still 3.9 and even higher at 0.6 $\mu$m. Are you really sure that Mie theory is not suitable?

**Page 15, line 11**: "...adequately." Please add a reference.

**Page 16, Fig. 10**: Why is $\nu_e$ indiced variability spread over such a large time period, especially for MSG-1 (6-11 UTC)?

**Page 16, line 13**: Is the assertion about the 3.9 $\mu$m phase function separation for different $\nu_e$ still valid if you rescale the plot (Fig. 11) as in Fig. 6? In principle, you should/could introduce a sort of phase separation index as in Cho et al. (2015) to quantitatively answer this question.

**Page 17, Fig. 14**: Which "part of the results" is expected? Why for "an optically moderately thick" cloud? Please explain.

**Page 18, line 5**: "wider size distributions are expected": please give a referece.

**Page 18, line 5-9**: The glory issue in the continental case should be investigated in the same detail as the marine one. It is not clear which features characterise the phase function at these higher angles (177-178°) that cannot be explained like for the 172° scattering angle in the marine case. By the way, a scattering angle plot for the continetal case should be presented.

**Page 20, Table 2**: Please add which cloud types (Sc, Cu...) have been investigated by Miles et al. (2000).

**Page 20, line 11**: What is meant by "In marine only clouds"?

**Page 20, line 22**: Please explain/rephrase "further emphasized".

**Technical corrections**

**Abstract**: The verbes should be in the present tense, e.g. "are analysed" instead of "were analysed".

**Page 1, line 2**: "... (LWP), which is a crucial component..." → marine low clouds are a crucial component, not LWP. Please rephrase.

**Page 1, line 9**: detection → observation.

**Page 1, line 15**: "different underlying surfaces" → please write land and ocean.

**Page 1, line 13**: "Cloud_cci" → please write ESA's Climate Change Initiative (essential climate variables related to clouds) or something like this.

**Page 1, line 15-16**: "more recent and advanced sensors provide high spatial and temporal resolution" → "... spatial and/or temporal resolution"

**Page 1, line 20**: I think that the CALIPSO/CALIOP lidar and CloudSat/CPR should be shortly mentioned here.

**Page 1, line 21**: "routinely retrieved from passive VIS-IR" → "routinely retrieved from e.g. passive VIS-IR" since also pure thermal algorithms exist, especially for cirrus clouds (e.g., Heidinger and Pavolonis, 2009; Holz et al., 2016; Minnis et al., 2016; Strandgren et al., 2017).

**Page 1, line 30**: "... biases reported..." → please cite already here the papers you mention below.

**Page 2, line 7**: "... not retrieved" → at this place the sentence "While under most retrieval circumstances..." on line 15 would fit particularly well.

**Page 4, line 12**: The height above the equator could be omitted.

[Figure]

**Page 4, line 19**: "diurnal basis" → "hourly basis".

**Page 5, line 6**: "12 spectral channels between" → "12 spectral channels in"

**Page 5, line 7**: Please mention also the HRV channel.

**Page 5, line 7**: Please introduce the CCP algorithm in Sect. 2.2 and not here.

**Page 5, line 9**: "near wavelength" → "centred at wavelength".

**Page 5, line 13-14**: Please mention the operational calibration slopes to have an idea about the differences in calibration.

**Page 6, line 20**: "contained within" → "filtered with"

**Page 7, line 1**: "approach" → "selection".

**Page 7, line 5**: Please rephrase "along with differences..." as "but the details of the phase functions for these scattering situations depend on the effective variance".

**Page 7, line 7**: 180 → 180°.

**Page 7, Table 1**: The rows in the third column are not aligned with the rows in the second column, please correct.

**Page 7, Fig. 2b**: Please add "scattering angle" to the x axis title.

**Page 8, line 3**: "equinox" → "vernal equinox".

**Page 8, line 7**: "spatial coverage" → "cloud cover".

**Page 8, line 9**: "more clouds" → "higher cloud cover".

**Page 8, line 9**: "over the continental region and less over the marine" → "over the continental and less over the marine region w.r.t. MSG-1".

**Page 8, line 11-12**: "their high spatial coverage with liquid clouds" → "the high liquid cloud cover".

**Page 9, Fig. 4a**: Please add "scattering angle" to the y axis title.

**Page 10, line 13**: "were based" → "are based".

**Page 10, line 24**: "showed" → "shows".

**Page 10, line 27**: "same" → "corresponding" or "appropriate".

**Page 10, line 27**: "the LUT now covers" → "the LUT for MSG-1 covers".

**Page 11, line 15**: "the phase function" → "the phase function at 1.6 $\mu$m".

**Page 11, line 16**: "the distance" → "the angular distance".

**Page 11, line 18**: "range" → "intensity".

**Page 11, Fig. 6**: Please plot larger ticks on the y axes.

**Page 15, line 6**: "increase" → "increases".

**Page 18, line 3**: "will affect" → "affects".

**Page 19, line 9**: "decreased flagged pixels" → "decreased numbers of flagged pixels".

**Mayer et al. (2015)**: Should read Mayer et al. (2004).

**Wood and Hartmann (2005)**: Should read Wood and Hartmann (2006).

**References**

Arduini, R. F., Minnis, P., Smith, W. L., Ayers, J. K., Khaiyer, K. K., and Heck, P.: Sensitivity of satellite-retrieved cloud properties to the effective variance of cloud droplet size distribution,

in: Proceedings of the 15th ARM Science Team Meeting, Atmospheric Radiation Measurement (ARM), 2005.

Cho, H.-M., Zhang, Z., Meyer, K., Lebsock, M., Platnick, S., Ackerman, A. S., Di Girolamo, L., C.-Labonnote, L., Cornet, C., Riedi, J., and Holz, R. E.: Frequency and causes of failed MODIS cloud property retrievals for liquid phase clouds over global oceans, J. Geophys. Res., 120, 4132–4154, https://doi.org/10.1002/2015JD023161, https://agupubs.onlinelibrary.wiley.com/doi/abs/10.1002/2015JD023161, 2015.

Heidinger, A. K. and Pavolonis, M. J.: Gazing at Cirrus Clouds for 25 Years through a Split Window. Part I: Methodology, J. Appl. Meteor. Climatol., 48, 1100–1116, https://doi.org/10.1175/2008JAMC1882.1, https://doi.org/10.1175/2008JAMC1882.1, 2009.

Holz, R. E., Platnick, S., Meyer, K., Vaughan, M., Heidinger, A., Yang, P., Wind, G., Dutcher, S., Ackerman, S., Amarasinghe, N., Nagle, F., and Wang, C.: Resolving ice cloud optical thickness biases between CALIOP and MODIS using infrared retrievals, Atmospheric Chemistry and Physics, 16, 5075–5090, https://doi.org/10.5194/acp-16-5075-2016, https://www.atmos-chem-phys.net/16/5075/2016/, 2016.

Mayer, B., Schröder, M., Preusker, R., and Schüller, L.: Remote sensing of water cloud droplet size distributions using the backscatter glory: a case study, Atmos. Chem. Phys., 4, 1255–1263, 2004.

Miles, N. L., Verlinde, J., and Clothiaux, E. E.: Cloud Droplet Size Distributions in Low-Level Stratiform Clouds, J. Atmos. Sci., 57, 295–311, https://doi.org/10.1175/1520-0469(2000)057<0295:CDSDIL>2.0.CO;2, https://doi.org/10.1175/1520-0469(2000)057<0295:CDSDIL>2.0.CO;2, 2000.

Minnis, P., Hong, G., Sun-Mack, S., Smith Jr., W. L., Chen, Y., and Miller, S. D.: Estimating nocturnal opaque ice cloud optical depth from MODIS multispectral infrared radiances using a neural network method, J. Geophys. Res., 121, 4907–4932, https://doi.org/10.1002/2015JD024456, https://agupubs.onlinelibrary.wiley.com/doi/abs/10.1002/2015JD024456, 2016.

Strandgren, J., Bugliaro, L., Sehnke, F., and Schröder, L.: Cirrus cloud retrieval with MSG/SEVIRI using artificial neural networks, Atmos. Meas. Tech., 10, 3547–3573, https://doi.org/10.5194/amt-10-3547-2017, https://www.atmos-meas-tech.net/10/3547/2017/, 2017.

---

## Author Comment (AC1) · 12 Apr 2019

We thank anonymous referee #1 for reviewing the manuscript and providing corrections and suggestions. Following are our point-by-point replies with the referee comments in italic. Page and line numbers refer to the marked-up revised manuscript, to be provided as a final response.

***General comments***

*The paper is well written and has an overall clear structure and figures. The topic is interesting and fits well to the aims and scopes of AMT. The authors identified two cloud features, bow and glory in the diurnal cycle of the cloud optical thickness and effective radius of stratocumulus clouds, which caused irregularities and could lead to misinterpretation by the user. The use of the two SEVIRI instruments onboard the Meteosat-8 and -10, which give the stereo perspective, is a great possibility to study these phenomena.*

*The sensitivity study focused only on the width of the droplet size distribution, which is important parameter and normally fixed for the cloud retrievals. The paper is valuable for people involved in cloud properties determination from image like SEVIRI measurements. The authors suggested two effective variance of the size distribution based on the sensitivity study. The whole study based only on two case studies. It is not shown that these two days are represented for a maritime and continual stratocumulus diurnal cycle, which should be included to make the results useful. Further the authors should discuss if this feature can be flagged for the user and include a suggestion how this cloud specific parameter could be used in standard retrievals as it mention in the abstract, but not discussed later.*

It is true that in our study we did not examine the level of representativeness of the selected cases for typical conditions in the corresponding marine and continental clouds. In the revised manuscript we comment on similarities and differences in the diurnal evolution of the marine case with the study by Seethala et al. (2018), which evaluated the typical diurnal cycle of marine Sc clouds over the south Atlantic. Our case appears typical for the diurnal evolution of $\tau$ in this region, and less typical for $r_e$. However, in order to perform this analysis a near overcast cloud deck during a whole day relatively close to the equinox was needed, which is not easy to find.

The cloud bow and cloud glory features are related to retrievals falling outside the LUT, and these are already flagged, as we clarify in the revised manuscript (page 18, lines 3-5). Hence the users can already use this information to minimize their effects.

*In principle the approach the authors present in their paper is very valuable and have great potential to understand cloud glory and cloud bow effects on the diurnal cycle of the cloud properties retrieved from satellite measurements. It is interesting and suitable for publication in AMT. However, I suggest the following revisions:*

***Abstract***
*Future climate data record is not mention or discussed in the paper at all, should be add in the discussion summary part.*

The relevant part of Sect. 4 (last paragraph) was expanded with reference to the CLAAS-3 and CLARA-A3 plans to update the $v_e$ value used, based on the findings of the present study. We also added more details on plans to attempt a retrieval of $v_e$ based on the information available in the glory time slot.

***Introduction***

*p. 3, line 4:*

*"Another issue in cloud optical properties retrieval, which relates to the cloud glory effects …" The citation from Mayer et al (2015) should come here already to motivate the sensitivity to ve. ( "While under most retrieval circumstances the sensitivity of τ and re to ve is low, this is not the case for special illumination geometries, as was shown e.g. in Mayer et al. (2015) for the cloud glory conditions.")*

This sentence was moved based on the referee's suggestion (page 3, lines 10-11).

*p. 3, line 24.*
*This is inconsistent to the abstract the authors mentioned: " .. over different underlying surfaces (ocean/land) .." and here "..over the southeast Atlantic and one characteristic day"*

We clarified this part by mentioning both the studied regions (page 3, lines 29-30).

*p.3 line 26-28.*
*Should be move to the summary: " While … properties."*

This sentence was moved to the beginning of Section 4.

***Data and Methodology***

*p.6 line 4-7. "It relies on … illumination conditions." This have not to repeat again. I would suggest to shorten this part. (see p.2 line 22-27.)*

The same piece of information was indeed repeated. We have shortened this part (page 6, lines 17-19).

*p. 8 line 2. Why only two days? Please discuss.*

Cloud glory and bow, and the ensuing irregularities in the retrievals, occur in specific time slots depending on the region. In order to combine both features in the analysis, we needed to narrow our selection of days to cases close to an equinox, when cloud glory features manifest. Hence, their study was necessarily limited to small areas and specific days. We have added this information in page 3, lines 26-29 of the revised manuscript. Irregularities due to cloud bow and glory conditions in the diurnal evolution of clouds will occur no matter how that diurnal cycle exactly looks. Hence, the diurnal evolution in the selected day(s) does not have to be representative of the specific cloud types and regions. In cases with variable cloudiness during the day, the impact of the cloud bow and glory features on the cloud property diurnal cycle may be blurred. Therefore, we searched for near-ideal cases in terms of cloud cover in a sufficiently large region during the full day.

*It would be useful to have RGB images for the two days and two satellite on a time slot.*

Compared to similar studies (e.g. Cho et al. (2015)), the areas under study are rather small (2°×2° based on the revised analysis) and covered to a large extent (more than 80%) by clouds. Hence we think that an RGB image would not add useful information, also considering that the analysis was mainly based on spatial averages.

***Results***

*For the spatially averaged reflectances it would be interesting to see the variation (max and min cloud glory and cloud bow effects) and the propagation to the CPP algorithm.*

Cloud bow and glory conditions lead to a momentary increase in reflectances in both the visible and the SWIR channels. This is depicted in Figs. 4c and 4e for the spatial averages, but it is true overall on a pixel basis. The histogram below shows an example of changes in reflectance at 1.6 μm from MSG-1, on 7/3/2017 over the southeastern Atlantic region, from 08:30 UTC, which is a "normal" time slot, to 08:45 UTC, which is affected by glory (see also Fig. 4e).

[Figure]

Propagation of this temporary "jump" to the CPP output is not straightforward to visualize, since the LUT, from which the output is retrieved, also changes shape through time. These changes ideally will compensate for the "jumps" in reflectances and lead to a diurnally smooth output. In practice, twists and folds in the LUTs (see e.g. Fig. 5a) lead pixels with extreme reflectance values to fall outside the LUT. These pixels are flagged, but retrievals are still performed, giving extreme output values. The retrieved $r_e$ values corresponding to the histograms above are shown below.

[Figure]

Obviously in this case the glory causes many pixels to lie above the LUT, leading to an unnatural peak in low $r_e$ values. These extreme retrievals lead to jumps when included in the spatial averages (see also Fig. 4f). A user should of course avoid using these flagged pixels. In our study, however, the aim is to analyze these cases and their effects, so their inclusion in the analysis was self-evident.

*Figure 4: The dotted and dashed line are hard to distinguish. For example, dotted dashed line and dotted line, should be better.*

We have added letters next to each vertical line, denoting either cloud bow (b) or glory (g).

*p. 14, line 13. Expression "more natural output" , please rephrase.*

This expression was rephrased (page 18, lines 2-3).

*p. 15, line 8. "this does not necessarily mean that the actual droplet size distribution is so narrow." Should be discussed how this could be verified in the conclusion.*

The purpose of this sentence was to state that the smoothness of the diurnal curve around the glory is not directly linked to the retrieval failures depicted in Fig. 9. However, this is not directly supported by the analysis, and it was removed from the revised version.

**Discussion and Summary**
*The authors should more discuss if this feature can be flagged for the user and include a suggestion how this cloud specific parameter could be used in standard retrievals as it mention in the abstract.*

The irregularities in cloud bow and cloud glory can already be avoided to some extent by the user, since they are associated with pixels falling outside the retrieval LUT, and these pixels are flagged (see also Fig. 9 and relevant discussion). Based on our findings, an updated value of $v_e$ will also be used in future retrievals of CLAAS and CLARA. This information was included in the last paragraph of the revised Sect. 4.

*p. 19, line 12. How often do these irregularities happen?*

Based on the way scattering angles change during a day, cloud bow irregularities will manifest twice per day in any region. Irregularities associated with cloud glory require high values of scattering angles. Due to the position of both satellites along the equator, these conditions are met in days close to the two equinoxes. This information was added in the revision (page 24, lines 16-18 and page 25, line 1).

*Should be critical discussed that the finding depends on the optical thickness of the cloud and the cloud types.*

The effect of $\tau$ on the cloud bow irregularities was already discussed in Sect. 3.4 (page 21, line 27 and page 22, lines 1-2). On the other hand, cloud glory irregularities on $\tau$ appear in both low and high $\tau$ cases.

**Typo**
*p.4 line 9: Data and Methodology*

Corrected.

---

## Author Comment (AC2) · 12 Apr 2019

We thank anonymous referee #2 for reviewing the manuscript and providing corrections and suggestions. Following are our point-by-point replies with the referee comments in italic. Page and line numbers refer to the marked-up revised manuscript, to be provided as a final response.

*Figures 6 and 11 show phase functions at intervals of angles of cloud bow and glory. It is seen that the phase functions are very sensitive to the effective size and to the effective variance. The reflectance of a cloud as measured from satellite at the top-of-atmosphere should be less sensitive because of multiple scattering. It would be instructive to provide figures of angular dependence of the reflectance as additional figures along with Figs 6 and 11. Additional figures can be done for one typical value of the cloud optical thickness, say 8, using the LUTs created by the authors. One observation geometry, say MSG-3 and the region west of the African coast, will be enough. Some of the Sun-satellite geometry angles can be constant and correspond to the case of March 7, 2017. The geometry angles should vary so that the intervals of scattering angles correspond to the intervals angles of Figs 6 and 11.*

The analysis proposed by the reviewer was included in our results (page 14, lines 2-12). We have selected the southeastern Atlantic region and the MSG-1 observation geometry, whereby both cloud bow and glory effects are apparent in the reflectances (Figs. 4c and 4e), during the same day (March 7, 2017). For each time slot, the viewing and illumination geometry and the effective radius were those calculated from the spatial averages of the actual retrievals, while three values of cloud optical thickness were examined: $\tau = 1$ (very thin cloud), $\tau = 8$ (close to the average retrieved value) and $\tau = 30$ (thick cloud). Using the LUT with $v_e = 0.15$, we plotted the reflectances in the 0.6 μm channel corresponding to these cases. The results clearly show how an increased $\tau$ increases the reflectance measured by the satellite sensor. They also confirm that the cloud glory and cloud bow reflectance magnitudes relative to non-glory and non-bow time slots are not affected by $\tau$, which was already mentioned in the manuscript for the cloud bow case. As Mayer et al. (2004) nicely described it: "The glory structure sits on top of a multiple-scattering background which of course depends on optical thickness".

*Figures 12 and 13 show retrievals results for 0.6 μm - 1.6 μm channels and 0.6 μm - 3.9 μm channels separately. There is difference in the retrieved values, especially for the effective size. Sensitivity to the effective variance is seen as well. The authors should discuss how those properties can be used to estimate the effective variance.*

Possible ways to estimate the effective variance are discussed in the last paragraph of Sect. 4 in the revised manuscript. They are not based, however, on the differences in retrievals using the two different spectral pairs, as the reviewer suggests. As we discuss in the marine case, where these differences are also present, they should be attributed to the different penetration depth of the 1.6 μm and the 3.9 μm wavelengths (page 19, lines 12-14), and possible shortcomings in the treatment of the 3.9 μm channel (page 21, lines 11-12).

*Technical corrections.*

*Page 3 line 16 Replace Mayer et al. (2015) by Mayer et al. (2004).*

Done.

*The dotted and dashed lines are hardly distinguished in figures. I would prefer to see straight vertical lines of different color.*

We have added letters next to each vertical line, denoting either cloud bow (b) or glory (g).

---

## Author Comment (AC3) · 12 Apr 2019

We thank anonymous referee #3 for reviewing the manuscript and providing corrections and suggestions. Following are our point-by-point replies with the referee comments in italic. Page and line numbers refer to the marked-up revised manuscript, to be provided as a final response.

***Specific comments***

*The paper combines methodologies mainly from Arduini et al. (2005) and Cho et al. (2015) with the results of Mayer et al. (2004).*

*The main deficiencies of the paper:*

*1. The authors talk about the cloudbow effect for scattering angles close to 140° (page 2, line 32) and the "collapse" of the phase function at a scattering angle of 132° and cite to this end the results of Cho et al. (2015) about retrieval failures for MODIS. Similarly, they cite this paper for the glory effect that also leads to retrieval failures for MODIS, but they also mention the work by Mayer et al. (2004) that uses the feature of the liquid cloud phase function around the backscatter direction to retrieve optical thickness, effective radius and effective variance of the particle size distribution. I think that in this paper two different effects are mixed together: on one side, the "bumps" in the phase function observed at the bow and glory geometries; on the other side the reduced sensitivity of the observations to effective radius due to the "collapse" of the phase function close to the same observation geometries.*
*An increased phase function intensity for particular scattering angles has as a consequence "stripes" of higher reflectivity for these scattering angles (as observed by Mayer et al. (2004) in the glory region) that, on their turn, could have as a consequence artificial "stripes" of higher optical thickness when the underlying LUT in the Nakajima-King retrievals does not consider these features (e.g. because of an unsufficient angular sampling of the phase function).*
*The "collapse" of the phase function instead produces retrieval failures due to the fact that the LUTs are narrower, especially for thin clouds, such that the retrieval is more prone to failure because the retrieval/radiative transfer/calibration uncertainties can "push" the observation out of the LUT more easily (cited more or less literally from Cho et al. (2015)). However, this collapse takes place in this paper (Fig. 5) at 132° and is the cloudbow effect found in Cho et al. (2015). The authors explain this effect in a nice way, but they assert that "in the cloud bow time slots retrievals are rather normal, with big differences occurring in $r_e$ for smaller scattering angles, namely close to 132°" (page 10, line 15-16). Thus, they mix up these two aspects that shall be separated clearly in the revised version of the manuscript. Correspondingly, it shall also be considered to plot the "collapse" angles instead of the cloudbow angles.*

It is true that the cloud bow and glory, on one hand, and the collapse of the phase functions, on the other, are two different effects, which we did not intend to mix. These effects are examined together in this study because they both cause irregularities in the retrievals. However, we think that mentioning that the effects of "collapse" occur "near" the cloud bow or glory, which we already mention in the first manuscript version (page 3, line 20), helps making their description simpler. In the revised manuscript, we try to clarify this further (page 3, lines 1).

While we have verified that our LUTs adequately consider the increased phase function intensity for these features, at least regarding the angular sampling, we also show that this phase function intensity is highly sensitive to the value of the effective variance (Fig. 6b), which has to be assumed. This sensitivity can explain the corresponding differences reported in cloud optical thickness for time slots around glory conditions, as we show e.g. in Figs. 7 and 8.

The problem caused by the collapse of the phase functions is indeed different, but it occurs *close to* the cloud bow, which is a "bump" in the phase function, as the reviewer describes. Hence, we don't understand why the statement in page 10, lines 15-16 of the initial manuscript is problematic. We think that it clearly separates the cases of cloud bow from those of scattering angles where the "collapse" occurs. Although Cho et al. (2015) describe similar failures as a "cloud bow effect", they clearly show their relation to the scattering angles where the phase functions collapse, and not to the local maximum in the phase function intensity, which defines the cloud bow. Both in the initial and the revised manuscript, we try to be specific regarding this discrimination, and a similar one regarding the cloud glory (see also our next reply), by using regularly the terms "near the cloud bow" and "around the cloud glory". However, starting the analysis with the observed reflectances, (Fig. 4c and e), we opted to plot the cloud bow angles, where reflectances peak locally.

*As far as the glory is concerned, I would suggest not to talk about maximum scattering angles and label them as "glory" (page 18, line 6) but to check the behavior of the phase function for angles larger than say 170° and then argument whether a particular effect on the cloud retrieval is expected at all (and in case which effect) or not. In this sense I think that also the information about the 0.6 µm phase function used in the argumentation on page 18, line 5-10 should be shown. Furthermore, the same effect in this scattering angle range (glory) might be expected as for the cloudbow as is explained in Cho et al. (2015). The "collapse" of the phase function mentioned at 132° is the most clear one, but Cho et al. (2015) identify further angles where this reduced sensitivity is also observed: they find 133°, 142° and 177° for their MODIS example. Even if at slightly different angles, this effect is present in this manuscript as well (see Fig. 5c and Fig. 6a) and could be the reason for the irregularities investigated. In particular, the 177° angle is a "glory angle" such that retrieval failures in this scattering angle range might also be traced back to a "phase function collapse".*
*These aspects shall be addressed in more clarity and the analysis adapted.*

It is true that the "maximum scattering angles" do not always coincide with a local maximum in the phase function intensity, which defines glory conditions, and this part was corrected accordingly. Nevertheless, the difference in scattering angles between adjacent 15-min time slots is of the order of 3°, ensuring that glory conditions will occur (perhaps more than once) close to this maximum. This probably explains also the difference in the shape of irregularities around the glory between MSG-1 and MSG-3 (see e.g. Fig. 8). Plotting the maximum scattering angles was rather selected as a common reference to guide the eye to the "center" of the period when glory effects occur. We clarify this point in page 11, lines 10-12 of the revised manuscript.

The referee's suggestion, namely to first check the behavior of the phase function for large angles and then argue whether a particular effect should be expected, is not feasible in our case, since it would imply that we already know the exact shape of the phase function, which depends on the effective variance.

Information about the 0.6 µm phase function is now included in Fig. 6b.

Following the referee's suggestion, in the revised manuscript we repeat the approach followed by Cho et al. (2015) and calculate separation indices for every group of phase functions we examine, in terms of both different $r_e$ values and different $v_e$ values (page 13, lines 20-30). This is a nice way to visualize and quantify the effect of "collapsing" phase functions on the retrieval process and helps in the interpretation of corresponding results. Our results verify the presence of "collapsing" conditions in the abovementioned scattering angles. It should be pointed out, however, that the 177° is not always a "glory angle", since it depends on the wavelength examined.

*2. To observe the effects described above I find the approach of using two fixed regions where the results of τ and $r_e$ are averaged (I think at least they are averaged: page 9 line 3-4 is not completely clear in this respect) to be not optimal since it might wash out the effects. In fact, all these "ripples" in the phase function take place in few degrees such that the approach of using regions with 5°×5° or 4°×4° size may complicate the identification and explanation of the related retrieval effects. In this sense, do for instance all pixels at 9 UTC in Fig. 4 have maximum scattering angle? I think that it should be discussed how strong the scattering angle can vary inside a SEVIRI pixel and inside the regions investigated in order to assess whether the expected effects can be identified or to what extent they are weakened by the averaging procedure. Furthermore, for a clearer illustration of the results I think that an additional picture showing one area (i.e. a 2D plot in latitude and longitude) in the cloudbow and/or glory slot would help interpreting the results (similarly to Fig. 10 in Cho et al. (2015)).*

The results in Fig. 4 are averaged, as are the reflectances. We clarify this point in page 10, lines 11-13 of the revised manuscript. The range of scattering angle values for each satellite and time slot was also quantified. Typical ranges for the southern Atlantic region are 0.91°±0.11° and 1.01°±0.11° for MSG-1 and MSG-3, respectively. It was also verified that the maximum scattering angle occurs in the same time slot for every pixel in both MSG-1 and MSG-3. While these results ensure no complication due to the size of the areas selected or the 15-min frequency of observations, we also examined averaging in smaller areas, based on the reviewer's concerns. We found that using 2°×2° areas we could still identify the effects under study, without increasing significantly the variability between time slots, which could compromise our results. However, the area covered with liquid clouds was significantly increased (Fig. 3 of the revised manuscript), and typical variations of scattering angle values within a time slot dropped to about 0.4°, providing a better correspondence between averaged retrievals and phase function characteristics. This is highlighted in page 11, lines 11-13 and page 12, lines 1-2 of the revised manuscript. Hence, we decided to repeat the analysis based on this reduced (2°×2°) area size.

However, the study areas are much smaller compared to the area shown in Fig. 10 of Cho et al. (2015) (~20°×20°), and uniformly covered with liquid clouds. Hence, adding a picture showing one of the areas would not provide any additional help in the interpretation of the results.

*3. Fig. 6b shows that there are angles of reduced sensitivity to the effective variance while most of the phase function shows a clear dependence on $v_e$ in this angle range. Can you see this "collapse" of the phase function w.r.t. $v_e$ in Fig. 7? Does this dependency of the phase function on $v_e$ extend to other scattering angles as well? Looking at Fig. 2 it would be interesting to shortly explain where an effect due to the dependence of the phase function on $v_e$ can be expected.*

A collapse of the phase function with respect to $r_e$, shown e.g. in Figs. 5c and 6a, leads to an irregularity in the $r_e$ retrieval (e.g. Figs. 8c and 8d near the cloud bow angle), independently of the $v_e$ value, which is selected *a priori*. This happens because phase functions collapse close to the cloud bow angular region no matter the $v_e$ value selected. Correspondingly, we should expect a reduced sensitivity in an attempt to retrieve $v_e$ in angles where phase functions of Fig. 6b collapse. Hence, the effect of the collapses in Fig. 6b cannot be seen in the results of Fig. 7. The irregularities in the $r_e$ retrieval around the maximum scattering angles (Figs. 7b, 8c and 8d), occurring especially for larger $v_e$ values, are rather associated with increased "collapsing" of the phase functions in the backscattering region for larger $v_e$ values, which leads to higher separation index values and hence more points falling outside the LUT.

*4. It has never been clearly stated in the manuscript whether the "flagged pixels" (e.g. page 14, line 14) contribute to the plots (e.g. Fig. 4). However, page 10 line 24-25 suggests that the τ - $r_e$ results for the*

*flagged pixels are used for the statistics (i.e. the diurnal cycles). From my point of view, these flagged pixels correspond to the failures investigated in Cho et al. (2015) and thus should not be used.*
*Similarly, do you show in Fig. 4c,4e the mean reflectance of the box or the mean cloud reflectance? This uncertainty arises from the phrasing "used as input to the CPP algorithm" (page 9, line 2) which seems to imply that you mean the cloudy pixels alone, since no retrieval is run for cloudfree pixels, I suppose. And what about the scattering angles/optical thickness/effective radius?*

"Flagged pixels", i.e. those falling outside the LUT, are indeed used in the diurnal plots. The purpose of providing flags in a data record is of course to inform the user on possible failures and reliability issues, and in that sense we agree with the reviewer that they should be excluded from a study using these data. In the present study, however, our purpose is to highlight the irregularities caused by these failures, and investigate their origin and ways to reduce their effects. Furthermore, exclusion of these pixels would cause gaps in the diurnal variation, or at least not directly comparable adjacent time slots, since flagged pixels cover large parts of the study areas in the specific time slots studied (Fig. 9).

The mean reflectance in Figs. 4c and 4e are mean liquid cloud reflectance, to be directly comparable with the CPP retrievals, which are also averaged over liquid clouds only. Scattering angles are averaged over the entire area. The almost complete coverage of the study areas with liquid clouds ensures that possible discrepancies are minimized. This is especially true for the revised results (see also revised Fig. 3). All these points are clarified in the revised manuscript (page 10, lines 11-13 and page 10, lines 15-16).

*5. In Fig. 4, the cloud glory regions (red and black) show different behaviours: the red one (MSG-1) shows a strong irregularity made up of two strong local minima and one local maximum in the optical thickness plot (d), while the black one (MSG-3) shows a weak local mimimum alone. Please explain why there is this difference.*

This difference should be attributed to the different scattering angles from the two satellites. The small differences, combined with the high sensitivity of the scattering phase functions in the backscattering directions, led to these different behaviors.

*6. I am missing an overall discussion about the plausibility of the retrieved diurnal cycles. This would increase also the plausibility of the investigations shown in the entire paper. For instance:*

- *Can one expect that marine Sc has an almost flat $r_e$ diurnal cycle (Fig. 4) while the optical thickness is decreasing strongly, a hint that the thermodynamic conditions the clouds are developing in are changing during the course of the day?*

  It is true that the flat $r_e$ diurnal cycle, combined with the strongly decreasing $\tau$, is not the average behavior of the marine Sc, where a decrease in $r_e$ would typically be expected (see e.g. Seethala et al., 2018). However, this is an one-day case, hence not necessarily representative of the average. We discuss this point in page 10, lines 14-15 of the revised manuscript.

- *The diurnal cycles in Fig. 4 and Fig. 10 differ: the 3.9 µm retrieval produces a lower $\tau$ and a lower $r_e$, although in an adiabatic environment one would expect higher $r_e$ at the cloud top, where the 3.9 µm retrieval is more sensitive. If you think that "subpixels fractions of open water" (page 17 line 9-10) are the reason for this, you might take a look at the HRV channel, if it is available over these regions, for a first check about this hypothesis. What is the uncertainty of the 3.9 µm retrieval, which should be higher than the one for the 1.6 µm?*

The study area is very homogeneously covered with clouds (and even more so is the adjusted 2°×2° area), and therefore the presence of subpixel fractions of open water is highly unlikely. We believe that potential imperfections in the treatment of the 3.9 µm channel (e.g., calibration, atmospheric absorption) are the most likely explanation for the differences between the 1.6 µm and 3.9 µm retrievals.

The calculated uncertainty in the 3.9 µm retrievals is overall somewhat lower than in the 1.6 µm retrieval. The probable explanation for this is that presently no uncertainty due to atmospheric absorption and the thermal contribution in the 3.9 µm retrievals is included in the calculations.

- *MSG-1 and MSG-3 in Fig. 10 provide different diurnal cycles of $r_e$. While MSG-1 seems to observe a decrease in re at around 9 UTC, MSG-3 yields an increase and a decrease afterwards.*

The decrease in $r_e$ retrieved from MSG-1 at around 9 UTC, also apparent based on the new results, is due to the glory conditions affecting the retrieval. Examining non-affected time slots, it can be seen that MSG-1 and MSG-3 retrievals are quite similar, with increasing $r_e$ before 9 UTC and slightly decreasing after 10 UTC.

- *In Fig. 12 MSG-1 and MSG-3 also provide very different diurnal cycles: not only the absolute values but also the variations in time are different, both in τ as well as in $r_e$. How can this be explained? Such strong differences preclude of course the use of simultaneous observations of the two satellites, both for physical/meteorological investigations of cloud properties and for the purpose of the present paper.*

It is true that there are many differences between MSG-1 and MSG-3 retrievals over the continental region, based also on the smaller study area of the revised manuscript. Although we are not certain on the reasons behind these differences, they are probably related to specific cloud types and 3D effects, since no similar issues were found in the marine case. These differences were not investigated further, since they don't actually preclude the separate use of the two sensors for the purpose of the present study: the simultaneous usage of the two satellites, which was based on the assumption that retrievals are the same, was dropped earlier in the study for practical reasons (see page 15, lines 9-10). The differences between MSG-1 and MSG-3 should indeed be of high concern if the previous assumption was required, and we highlight this point on page 15, lines 10-11 of the revised manuscript. In Section 3.4, however, they are only analyzed in parallel, to highlight the same effects based on different illumination and viewing conditions. However, since this point is important, we also included a small discussion acknowledging this issue (page 22, lines 12-16).

*7. The continental case is said to be "not directly comparable with the marine Sc case" (page 18, line 5). If this case is shown, and I think the paper benefits from this since it shows a cloud with higher optical thickness and much higher effective radius, it should be done in more detail: see my comment above about the phase function and the explanation of the diurnal cycles. Further aspects that are not completely clear for this case are the fact that τ shows a dependency on $v_e$ while $r_e$ shows none in the glory geometry, and the short temporal displacement between the small local minima in τ for the cloudbow and the local maxima in $r_e$ (Fig. 12).*

Section 3.4 was expanded in the revised manuscript based on the aspects raised by the reviewer, as well as on the additional analyses of uncertainties and separation indices. Our arguments regarding differences between $\tau$ and $r_e$ under glory conditions were based primarily on the characteristics of the 0.6 μm and 1.6 μm phase functions.

*8. Retrieval results still seem to show a relatively small variability w.r.t. $v_e$. Is this sensitivity to $v_e$ comparable to the retrieval uncertainties or retrieval errors or is it larger?*

While it is not clear to which retrieval results the reviewer refers to, it is indeed crucial for the validity of our conclusions to compare the variability w.r.t $v_e$ with retrieval uncertainties. For this reason we used the methodology described in Stengel et al. (2017) to propagate the level 2 retrieval uncertainties to the spatially averaged values used in this study. We describe the process used in page 8, lines 8-14 and page 9, lines 1-2. In the Results section of the revised manuscript, it is also explicitly mentioned if the $v_e$ variability is smaller or larger than the propagated uncertainty.

*9. The manuscript demonstrates that particular geometries like the cloudbow and the glory can lead to biased optical properties, but what would you propose in order to reduce this bias, keeping in mind that in 15 minutes (from one slot to the next) also cloud physics can vary (the cloud can thin out, become thicker, its particle size distribution can change...)?*

Based on our results, we propose that using more appropriate values of effective variance can help reducing the biases associated with the cloud glory geometry (page 26, lines 13-14). While cloud properties indeed vary within 15 minutes, this variability is not apparent in our spatially averaged analysis. On a pixel basis, this variability is expected to be more pronounced, but correcting the size distribution as we propose would still remove the additional irregularities.

*10. The paper should shortly discuss/mention at one place the reasons why the same retrieval from two satellites at the different locations could yield different results, apart because of glory and cloudbow. Here I think of shadow effects, partially cloudy pixels, cloud inhomogeneities, 3D radiation effects, surface BRDF, mixed phase clouds, misidentification of thin cirrus on top... This is the basis for the synergistic use of the two MSG spacecraft. Parts of this discussion are e.g. at page 8 line 8 and page 10 line 11.*

This is indeed an important part of the discussion that we included in page 4, lines 11-18.

*Further comments:*

**Title**: *Since the paper presents results for two selected days over two selected regions I recommend to add "A case study" somewhere to the title.*

While the paper indeed examines two selected days, we would hesitate to characterize it as a "case study" for two main reasons: (a) any further averaging of data, either in space or in time, would render the cloud bow and glory phenomena impossible to study, since their effects are restricted in both space and time. Hence, this study cannot be conducted on a "bulk-data" basis. (b) The notion of "case study" usually refers to a phenomenon of particular interest, which manifests as a specific event (e.g. extreme weather event). However, the days studied here were chosen based only on specific criteria that were met. The phenomena studied occur around the globe on a daily basis.

**Abstract:** *Please mention that you analysed two days of data.*

Done.

**Figures:**

• *I suggest to merge Fig. 3a with Fig. 4 and Fig. 3b with Fig. 12.*

We opted to keep the figures separated, because we wanted to highlight from the beginning that both study regions met (and were selected based on) the high coverage with liquid clouds criterion. Furthermore, Fig. 4 would become too "busy" with such a merging.

• *Please use the same colors for MSG-1 and MSG-3 in all figures.*

The reviewer probably refers to Fig. 3, where colors were swapped. This is now corrected.

• *Please add the solar zenith angle to Fig. 4 in order to understand when $\theta_0$ is reaching 90°, i.e. sunrise and sunset. This might explain for instance the increasing reflectance at 0.6 µm in Fig. 4c and 4e.*

The solar zenith angle in these plots is always less than 84.3°. In fact, this is the threshold for CPP retrievals used in CLAAS-2 and applied throughout this study. This is mentioned in Table 1 but is further clarified in the revision (page 9, lines 20-21). Hence, sunrise and sunset cannot explain the increasing reflectances in Figs. 4c and 4e, and adding the solar zenith angle to Fig. 4 would not be useful. As explained in a later comment (page 10 of this document), the increasing reflectances are probably due to the high solar zenith angle combined with the higher viewing angle from MSG-1 compared to MSG-3. This difference in reflectance, however, is not repeated in the CPP output, showing that these conditions are accounted for in the retrieval.

• *I suggest to move Fig. 5c to Fig. 6.*

Figure 5c was placed next to the LUT diagrams of Figs. 5a and 5b to highlight the effect of the "collapse" of the phase functions at 132° on the corresponding LUT depicted in Fig. 5a. The phase functions of Fig. 6 focus on the backscattering directions and the relevant discussion refers to cloud glory. For this reason we think that it is better to keep Fig. 5c and Fig. 6 separated.

• *For all figures with glory and cloudbow: it would be helpful for the reader to write directly into the plot which vertical line is glory and which one is cloudbow.*

We have added letters next to each vertical line, denoting either cloud bow (b) or glory (g).

• *For all figures with diurnal cycles: it would be easier for the reader if each panel contained MSG-1 or MSG-3 somewhere to distinguish the satellites at a glance.*

Done.

• *Since only hours are used in the diurnal cycle plots I think that e.g. "05" or "5" would be better than "0500".*

Done.

*• Units should be expresses either as e.g. "Θ / degree" or "Θ [degree]" but not "Θ (degree)". Furthermore "Reflectance (0.6 µm)" should read "Reflectance at 0.6 µm" with no unit. Instead of "Hour (UTC)" I suggest "UTC hour".*

Done.

*• Fig. 5 is too small. Furthermore, it is probably not a "scatter plot" (page 10, line 25-26) but I guess a 2D histogram. In that case the colors should be explained as well.*

Indeed, the term "density plot" explains better what Figs. 5a and 5b show (page 13, lines 3). The colors are also explained in the revised figure caption.

*• Please add a (dotted) line at height 0 ($r_e$ or $\tau$ difference = 0) in Fig. 7.*

Done.

***Page 4, line 21****: I cannot believe that the MSG-1 satellite is moving so fast and so much (10° latitude in 24 h) around its subsatellite point. Please check this in more detail! This could have an important impact of the observation geometry.*

Sub-satellite point coordinates are available in the original MSG SEVIRI files metadata, while EUMETSAT has also warned users on this issue (see e.g. www.eumetsat.int/website/home/News/ConferencesandEvents/DAT_3647214.html). This deviation, however, should not be seen as an independent movement of the satellite around the nominal sub-satellite point, but rather as the satellite lying in an orbital plane inclined by about 5° compared to the equatorial plane. While the impact on the observation geometry can indeed be important, inclusion of this information in our retrievals actually ensures avoidance of possible misinterpretations.

***Page 6, line 21****: Why do you need three values of the surface albedo?*

The explanation for this is somewhat technical. The cloud reflectance can be written as the sum of the reflectance for a dark underlying surface and a term containing cloud transmittance, the hemispherical sky albedo for upwelling isotropic radiation, and the actual surface albedo. From radiative transfer simulations of the cloud reflectance at two particular surface albedos (chosen as 0.5 and 1 for numerical stability), the transmittance (function of zenith angle) and sky albedo can be determined. These are also stored in the look-up table and then allow the direct calculation of cloud reflectance for any value of the surface albedo. This procedure is described in the CM SAF CLAAS-2 ATBD, and a reference to this ATBD has been added in the manuscript (CM SAF, 2016a; page 7, lines 12).

***Page 6, line 24-27****: Please give a reference (or a short explanation) for the gas absorption correction and the thermal emission consideration.*

The gas absorption correction method is explained in the CM SAF CLAAS-2 ATBD. The thermal emission calculation is not covered by the CLAAS-2 ATBD because CLAAS-2 does not involve the 3.9 µm channel. However, the CPP algorithm has also been applied to AVHRR for the production of the CLARA-

A2 data record. In that context, the AVHRR 3.7 μm channel is used, and the consideration of thermal emission is covered in the corresponding ATBD, which has been added to the main text (page 7, line 10) and the reference list:

CM SAF: Algorithm Theoretical Basis Document, CM SAF Cloud, Albedo, Radiation data record, AVHRR-based, Edition 2 (CLARA-A2), Cloud Physical Products, EUMETSAT Satellite Application Facility on Climate Monitoring, SAF/CM/SMHI/ATBD/CPP_AHVRR issue 2.0, 19/08/2016, doi: 10.5676/EUM_SAF_CM/CLARA_AVHRR/V002, 2016b.

***Page 7, line 3-4****: The size distributions do not depend on wavelength (line 3), but the phase functions do (line 4), so please shift "for the visible wavelength (0.6 μm)" after "phase functions". Please correct also the caption of Fig. 2.*

Corrected.

***Page 7, line 4-5****: Please indicate in the text and/or in the figure where the cloudbow and glory features can be observed.*

Cloud bow and glory features manifest as peaks near 140° and in the backscattering direction, respectively. This is added in the revised manuscript (page 7, lines 20-21)

***Page 7, Table 1****: Is there such a set of LUTs for every value of the surface albedo mentioned in the text? Please explain this.*

No, as explained earlier, radiative transfer calculations for three values of the surface albedo are used to derive cloud transmittance and hemispherical sky albedo, which are stored in the LUT, and allow the calculation of the cloud reflectance for any albedo of the underlying surface.

***Page 8, line 4****: Please indicate in the text here and not only in the caption of Fig. 1 the details of the region coordinates.*

Added (page 9, lines 6-7).

***Page 8, line 5****: Please mention which quantity has been used to assess "uniformity" of the cloud deck.*

By "uniformity" we mean high degree of spatial coverage. We have rephrased accordingly (page 9, line 7).

***Page 8, line 8-10****: This argument, related to the different viewing conditions, also depends on the cloud field observed. If the Sc has dimensions that are anyway smaller than the spatial resolution of SEVIRI, only small differences might appear here.*

This is indeed the case in the updated results. Differences are larger in the continental case and only minor in the marine region (Fig. 3).

***Page 8, line 10****: What does it mean that MSG-1 detected "more ice clouds"? Are there ice clouds during these days? Do they contribute to the cloud cover shown in Fig. 3? Are there ice clouds that MSG-3 does not detect and contribute to the retrieval results (Fig. 4 onward)? Are ice clouds maybe one of the*

*reasons for the differences in cloud cover from MSG-1 and MSG-3? Which further factors might explain these differences?*

Examination of the retrieved cloud phase data showed that over the southeastern Atlantic between 12:30 and 17:00 UTC MSG-1 retrieved ice clouds covering 3-4% of the 5° × 5° study region, reaching 17% in 16:30 and 16:45 UTC. For the same time slots ice cloud cover from MSG-3 never exceeded 0.5%. Over the 4° × 4° continental region both satellites detected ice clouds covering between 10% and 20% from 14:00 UTC to 16:00 UTC and about 5% in other time slots. While these findings can explain results shown in Fig. 3, ice clouds did not contribute to later retrieval results, which were constrained to pixels where both satellites retrieved liquid clouds. Furthermore, in the revised manuscript, where both study regions are decreased to 2° × 2°, ice cloud coverage over the southeastern Atlantic for the same time slots from MSG-1 never exceeds 4%, while no ice cloud is detected from MSG-3. Over the 2° × 2° continental region, ice clouds never exceed 2% in either satellite retrieval. Hence, this statement was removed from the revised manuscript.

***Page 8, line 14***: *At this point of the manuscript it is not clear yet when the cloudbow and glory geometry occur, so please explain this in the text. Nevertheless, if you merge this figure with Fig. 4 or 12, as suggested above, this remark is superfluous.*

Our purpose here is not to highlight the cloud bow and glory time slots, but rather to mention that they are also included in the time range with high liquid cloud cover. This part was rephrased accordingly (page 9, lines 18).

***Page 8, line 21***: *In the MSG-1 curve in Fig. 3a there are discontinuities in the afternoon while the MSG-3 cloud cover is very smooth. Might they be caused by sunglint? Which possible effects might explain these differences otherwise?*

Figure 3 was updated based on the revised analysis and the reduced size of the study regions (2°×2°). Discontinuities in the MSG-1 curve are now much less pronounced. Sunglint conditions are included in CPP as a flag, and it was indeed found that these conditions were fulfilled for some pixels and time slots in late afternoon over the southeast Atlantic region with MSG-1. However, their effect on CPP retrievals would be significant only if clear-sky pixels were misinterpreted as cloudy. The good agreement between MSG-1 and MSG-3 cloud cover ensures that this is not the case here.

***Page 10, line 6-7***: *"with values increasing rapidly ..." ! Please explain.*

This increase in reflectances from MSG-1 in late afternoon, which is not present in MSG-3, should probably be attributed to a combination of the sun positioned low above the horizon and the viewing angle of MSG-1, which is larger than that of MSG-3. It should be noted however that, based on the CPP output, where no similar difference is found, these conditions are accounted for in the retrieval LUTs. This is clarified in page 12, lines 7-9 of the revised manuscript.

***Page 10, line 10***: *"different illumination conditions" ! please explain.*

Here we mean illumination *and viewing* conditions. We have rephrased accordingly (page 12, line 12).

***Page 10, line 27***: *Please mention the observation conditions that are shown here.*

The observation conditions ($\theta, \theta_0$ and $\Theta$) are given in the revised Fig. 5 (for MSG-1: $\theta = 43.4°$, $\theta_0 = 55.7°$, $\Theta = 86.1°$; for MSG-3: $\theta = 22.7°$, $\theta_0 = 55.7°$, $\Theta = 134.1°$).

*Page 11, line 1-3: This is an interesting point and also not obvious since the reflectance observed by the satellite is affected by single and by multiple scattering at the same time. Thus it is not trivial to find a signature of the single scattering properties in this quantity. Please consider mentioning this aspect in the text.*

We agree with the reviewer and we have included this aspect in the text. We have also added that this LUT characteristic affects optically thin clouds only, where single scattering prevails (page 13, lines 7-11).

*Page 11, line 9: Please quantify "thin".*

Based on the LUT in Fig. 5a, $\tau < 4$ would be a rough quantification (included in page 13, line 18).

*Page 11, line 15: Is this assertion from Mayer et al. (2004) who used reflectances at 753nm also valid for other wavelengths? I think you can cite your plots as well to explain this.*

This part was rephrased to reflect the referee's suggestion (page 14, lines 1-2).

*Page 12, line 9-11: Could you please explain what you mean with this sentence, in particular with "their differences"?*

In the case of $\tau$, where differences due to different size distributions appear only in cloud glory conditions (Fig. 7a), the sensitivity to $v_e$ is clear. In the case of $r_e$, however, we have differences in both glory but also large irregularities in cloud bow conditions, and the glory from one satellite almost coincides with the cloud bow from the other. Hence, it is difficult to discern the sensitivity to $v_e$. This part is rephrased in the revised manuscript (page 15, lines 8-10).

*Page 12, line 11: What is meant with "This is due..."?*

We mean the difficulty to discern the sensitivity to $v_e$. This sentence has been rephrased (see also our reply to the previous comment).

*Page 12, line 12-13: If you "give up" your synergistic approach of using MSG-1 and MSG-3 you might consider showing results from only one satellite. This would make the next figures "lighter" and you can eventually mention that these results are confirmed (not shown) by the other satellite.*

While the reviewer's suggestion would indeed make the presentation of results "lighter", we opted to continue the analysis with both satellites. The main reason is that including a second satellite adds cases with different scattering angles, which can contribute to the analysis.

*Page 13, Fig. 7: Why is the effect of the glory smaller for MSG-3?*

This difference should be attributed to the different maximum scattering angles: 176.4° for MSG-1 and 172.6° for MSG-3. An inspection of Fig. 6a, and especially the corresponding separation index

values (Fig. S3 of the supplement) shows that indeed MSG-1 would be more prone to failures, and thus irregularities, than MSG-3. This is explained in the revised manuscript (page 17, lines 12-15).

***Page 14, line 7****: Please specify "significant effect" and put it in relation to the uncertainty of the $r_e$ retrieval.*

The term "significant" here was not meant in a statistical sense, and was replaced by "strong" for more clarity. The retrieval uncertainties are analyzed in the revised manuscript as described in page 8, lines 8-14, and page 9, lines 1-2, and they are discussed in relation to our results throughout the manuscript.

***Page 14, line 8****: Please explain what you mean by "The effect on the glory is similar to the τ case".*

We mean that the effect of $v_e$ on $r_e$ in the glory is similar to the effect of $v_e$ on $τ$ in the glory. We have rephrased accordingly (page 17, lines 11-12).

***Page 14, line 14****: "which are flagged" as bad quality? As uncertain?*

As pixels where the pair of VIS and SWIR reflectances lies outside the Nakajima-King LUT. We have added this clarification (page 18, line 4).

***Page 15, line 10-11****: "… these distributions cannot capture the cloud glory adequately." Do you mean that such distributions do not show the glory effect or that Mie theory is not adequate for such distributions? The size parameter for 1 μm particles (small cloud droplet) at 1.6 μm is still 3.9 and even higher at 0.6 μm. Are you really sure that Mie theory is not suitable?*

We mean that in such distributions the cloud glory effect is much weaker, as can also be seen in Figs. 6b and 6c for phase functions at 0.6 μm and 1.6 μm, respectively. We have rephrased accordingly (page 18, line 19).

***Page 15, line 11****: "…adequately." Please add a reference.*

This part has been rephrased (see previous comment).

***Page 16, Fig. 10****: Why is $v_e$ indiced variability spread over such a large time period, especially for MSG-1 (6-11 UTC)?*

This spread should be attributed to the different characteristics of the 3.9 μm phase function in the backscattering directions. Specifically, glory features are wider compared to smaller wavelengths (see e.g. Figs. 11a and 6b), covering a larger range of scattering angles, hence the variability will be spread over a larger time period. This is added in the relevant description of Fig. 10 (page 20, lines 19-21).

***Page 16, line 13****: Is the assertion about the 3.9 μm phase function separation for different $v_e$ still valid if you rescale the plot (Fig. 11) as in Fig. 6? In principle, you should/could introduce a sort of phase separation index as in Cho et al. (2015) to quantitatively answer this question.*

Following the reviewer's suggestions, we introduced the Cho et al. (2015) phase separation index in our analysis. Index values at 172° scattering angle are indeed smaller in the 3.9 μm phase function compared to the 1.6 μm phase function (Figs. S5 and S7 of the supplement). This is also mentioned in the revision (page 20, lines 15-18).

*Page 17, Fig. 14: Which "part of the results" is expected? Why for "an optically moderately thick" cloud? Please explain.*

This sentence has been rephrased for clarity (page 21, lines 14-15).

*Page 18, line 5: "wider size distributions are expected": please give a reference.*

This expectation is based on the results of Miles et al. (2000). Added in page 22, line 4.

*Page 18, line 5-9: The glory issue in the continental case should be investigated in the same detail as the marine one. It is not clear which features characterise the phase function at these higher angles (177-178°) that cannot be explained like for the 172° scattering angle in the marine case. By the way, a scattering angle plot for the continental case should be presented.*

This part has been expanded. The relevant phase function is now shown (Fig. 6b) along with the respective separation index (Fig. S3). Scattering angles for the continental case are also shown in revised Fig. 12.

*Page 20, Table 2: Please add which cloud types (Sc, Cu...) have been investigated by Miles et al. (2000).*

Added in page 25, lines 22-24.

*Page 20, line 11: What is meant by "In marine only clouds"?*

We meant "marine Sc clouds". We have rephrased accordingly (page 25, lines 25).

*Page 20, line 22: Please explain/rephrase "further emphasized".*

Here we mean the fact that the conclusions drawn from the present study are similar to those from previous studies. We have rephrased this part for more clarity (page 26, lines 11-13).

***Technical corrections***

*Abstract: The verbs should be in the present tense, e.g. "are analysed" instead of "were analysed".*
Done.
*Page 1, line 2: "... (LWP), which is a crucial component..." ! marine low clouds are a crucial component, not LWP. Please rephrase.*
Rephrased.
*Page 1, line 9: detection ! observation.*
Changed.
*Page 1, line 15: "different underlying surfaces" ! please write land and ocean.*
Done.

***Page 1, line 13***: *"Cloud_cci"!please write ESA's Climate Change Initiative (essential climate variables related to clouds) or something like this.*

        Done.

***Page 1, line 15-16***: *"more recent and advanced sensors provide high spatial and temporal resolution" ! "... spatial and/or temporal resolution"*

        Changed.

***Page 1, line 20***: *I think that the CALIPSO/CALIOP lidar and CloudSat/CPR should be shortly mentioned here.*

        We did not mention CALIPSO/CALIOP and CloudSat/CPR since the focus is on passive imagers only.

***Page 1, line 21***: *"routinely retrieved from passive VIS-IR" ! "routinely retrieved from e.g. passive VIS-IR" since also pure thermal algorithms exist, especially for cirrus clouds (e.g., Heidinger and Pavolonis, 2009; Holz et al., 2016; Minnis et al., 2016; Strandgren et al., 2017).*

        We focus on the VIS-IR methods here. It is not relevant that cloud optical and microphysical properties are also retrieved from other wavelengths.

***Page 1, line 30***: *"... biases reported..." ! please cite already here the papers you mention below.*

        Done.

***Page 2, line 7***: *"... not retrieved" ! at this place the sentence "While under most retrieval circumstances..." on line 15 would fit particularly well.*

        The sentence was moved.

***Page 4, line 12***: *The height above the equator could be omitted.*

        Done.

***Page 4, line 19***: *"diurnal basis" ! "hourly basis".*

        Changed.

***Page 5, line 6***: *"12 spectral channels between" ! "12 spectral channels in"*

        Changed.

***Page 5, line 7***: *Please mention also the HRV channel.*

        Done.

***Page 5, line 7***: *Please introduce the CCP algorithm in Sect. 2.2 and not here.*

        This part was moved to the beginning of Sect. 2.2.

***Page 5, line 9***: *"near wavelength" ! "centred at wavelength".*

        We really mean 'near'. The central wavelength is not 0.6 $\mu$m but 0.635 $\mu$m. For clarity we mention the exact central wavelengths and the approximate values used further on in the paper in the revised manuscript (page 6, line 11).

***Page 5, line 13-14***: *Please mention the operational calibration slopes to have an idea about the differences in calibration.*

        We introduced the following phrase: 'These values can be compared with the corresponding operational calibration slopes of 0.0241, 0.0233, 0.0209, and 0.0236 mW m$^{-2}$ sr$^{-1}$ (cm$^{-1}$)$^{-1}$ , respectively.' (page 6, lines 6-7).

***Page 6, line 20***: *"contained within" ! "filtered with"*

        Changed.

***Page 7, line 1***: *"approach" ! "selection".*

        Changed.

***Page 7, line 5***: *Please rephrase "along with differences..." as "but the details of the phase functions for these scattering situations depend on the effective variance".*

        Rephrased.

***Page 7, line 7***: *180 ! 180°.*

        Corrected.

***Page 7, Table 1***: *The rows in the third column are not aligned with the rows in the second column, please correct.*

      Corrected.

***Page 7, Fig. 2b***: *Please add "scattering angle" to the x axis title.*

      Added.

***Page 8, line 3***: *"equinox" ! "vernal equinox".*

      Added.

***Page 8, line 7***: *"spatial coverage" ! "cloud cover".*

      Replaced.

***Page 8, line 9***: *"more clouds" ! "higher cloud cover".*

      Replaced.

***Page 8, line 9***: *"over the continental region and less over the marine" ! "over the continental and less over the marine region w.r.t. MSG-1".*

      Rephrased.

***Page 8, line 11-12***: *"their high spatial coverage with liquid clouds" ! "the high liquid cloud cover".*

      Rephrased.

***Page 9, Fig. 4a***: *Please add "scattering angle" to the y axis title.*

      Added.

***Page 10, line 13***: *"were based" ! "are based".*

      Changed.

***Page 10, line 24***: *"showed" ! "shows".*

      Changed.

***Page 10, line 27***: *"same" ! "corresponding" or "appropriate".*

      Changed.

***Page 10, line 27***: *"the LUT now covers" ! "the LUT for MSG-1 covers".*

      Rephrased.

***Page 11, line 15***: *"the phase function" ! "the phase function at 1.6 μm".*

      Added.

***Page 11, line 16***: *"the distance" ! "the angular distance".*

      Added.

***Page 11, line 18***: *"range" ! "intensity".*

      Replaced.

***Page 11, Fig. 6***: *Please plot larger ticks on the y axes.*

      Done.

***Page 15, line 6***: *"increase" ! "increases".*

      Corrected.

***Page 18, line 3***: *"will affect" ! "affects".*

      Changed.

***Page 19, line 9***: *"decreased flagged pixels" ! "decreased numbers of flagged pixels".*

      Changed.

***Mayer et al. (2015)***: *Should read Mayer et al. (2004).*

      Corrected.

***Wood and Hartmann (2005)***: *Should read Wood and Hartmann (2006).*

      Corrected.

**References**

Seethala, C., Meirink, J. F., Horváth, Á., Bennartz, R., and Roebeling, R.: Evaluating the diurnal cycle of South Atlantic stratocumulus clouds as observed by MSG SEVIRI, Atmos. Chem. Phys., 18, 13283-13304, doi: 10.5194/acp-18-13283-2018, 2018.

Stengel, M., Stapelberg, S., Sus, O., Schlundt, C., Poulsen, C., Thomas, G., Christensen, M., Carbajal Henken, C., Preusker, R., Fischer, J., Devasthale, A., Willén, U., Karlsson, K.-G., McGarragh, G. R., Proud, S., Povey, A. C., Grainger, R. G., Meirink, J. F., Feofilov, A., Bennartz, R., Bojanowski, J. S., and Hollmann, R.: Cloud property datasets retrieved from AVHRR, MODIS, AATSR and MERIS in the framework of the Cloud_cci project, Earth Syst. Sci. Data, 9, 881-904, doi: 10.5194/essd-9-881-2017, 2017.